# Motion-Residual Conflict-Aware Time Reversal for Generative Inbetweening

Zhenbang Zhang [* 1]  Zihui Cui [* 1]  Haythem El-Messiry [2]  Renmin Han [3]  Zhiqiang Xu [1]

## Abstract

Image-to-video (I2V) diffusion models have recently made generative inbetweening a practical reality by synthesizing semantically plausible intermediate frames between two keyframes. Among them, inference-time sampling schemes that re-use large pre-trained I2V backbones without any additional training are especially attractive. Yet current methods frequently exhibit temporal inconsistency and artifacts such as ghosting or reverse motion. A key reason is that the two trajectories are driven by distinct motion priors, each inherited from its own conditioning frame, and are simply stitched together without explicitly reconciling these priors. We introduce Motion-Residual Conflict-Aware Time Reversal (MR-CATR), an inference-time sampling framework that aligns conflicting motion priors instead of discarding one of them or collapsing to a single start-conditioned prior. MR-CATR first derives a motion-residual–based direction from the forward path, combined with an end-conditioned residual to form a consensus motion axis. This design suppresses bidirectional motion conflicts while still allowing end-frame information to refine the trajectory and enforce endpoint consistency. MR-CATR can be seamlessly integrated into existing time-reversal samplers without changing model parameters. Experiments on generative inbetweening benchmarks show that our method produces videos with smoother motion, fewer artifacts, and consistently better quantitative scores and user preferences than prior strategies. Code is available at https://github.com/zhangzhenbang2021/MR-CATR.git.

---

[*]Equal contribution  [1]MBZUAI, Abu Dhabi, United Arab Emirates [2]Canadian University Dubai, Dubai, United Arab Emirates [3]Shandong University, Shandong, China. Correspondence to: Renmin Han <hanrenmin@sdu.edu.cn>, Zhiqiang Xu <zhiqiang.xu@mbzuai.ac.ae>.

*Proceedings of the 43$^{rd}$ International Conference on Machine Learning*, Seoul, South Korea. PMLR 306, 2026. Copyright 2026 by the author(s).

## 1. Introduction

Diffusion models have recently reshaped image and video generation, offering strong fidelity and diversity across a wide range of visual tasks. In particular, image-to-video (I2V) diffusion models (Blattmann et al., 2023; Huang et al., 2024a; Shi et al., 2024; Wang et al., 2025) are able to synthesize temporally coherent video clips from a single input frame. While existing I2V diffusion models excel at generating videos from one conditioning frame, they are not directly designed for bounded generation, where both the start and end frames must be taken into account simultaneously.

To handle such dual constraints, recent work has adopted time reversal sampling, which maintains two denoising trajectories: a forward path conditioned on the start frame and a backward path conditioned on the end frame. Depending on how these trajectories are combined, current approaches can be grouped into parallel and sequential schemes. Parallel methods (Feng et al., 2024; Zhu et al., 2025; Wang et al., 2025) denoise the two paths at each diffusion step and then linearly blend the two states to obtain the next iterate. In contrast, sequential strategies (Yang et al., 2024) first follow the start-conditioned path, insert a re-noising step, and then continue denoising along the end-conditioned path. Despite their effectiveness, these strategies primarily focus on how to connect two temporal paths, rather than how to reconcile the motion priors that generate them.

Recent work Motion Prior Distillation (MPD) (Jeon et al., 2026) observes that, due to the forward-training bias of I2V models that are optimized to predict future frames given a single image, the end-conditioned backward path tends to produce *forward-looking* sequences rather than faithfully reconstructing the past. As a result, the forward and backward paths may predict incompatible motions for the same scene, leading to ghosting artifacts and temporal inconsistency. As shown in Figure 2 (c), MPD explicitly distills the forward motion residual into the backward path and reconstructs end-aligned denoised estimates solely from the start-induced residuals. However, the backward path no longer carries its own motion information from the final frame, making the sampler highly dependent on the quality of the forward trajectory. Moreover, when the ending keyframe introduces new objects or the scene undergoes large-scale rearrangement, enforcing a globally unified start-

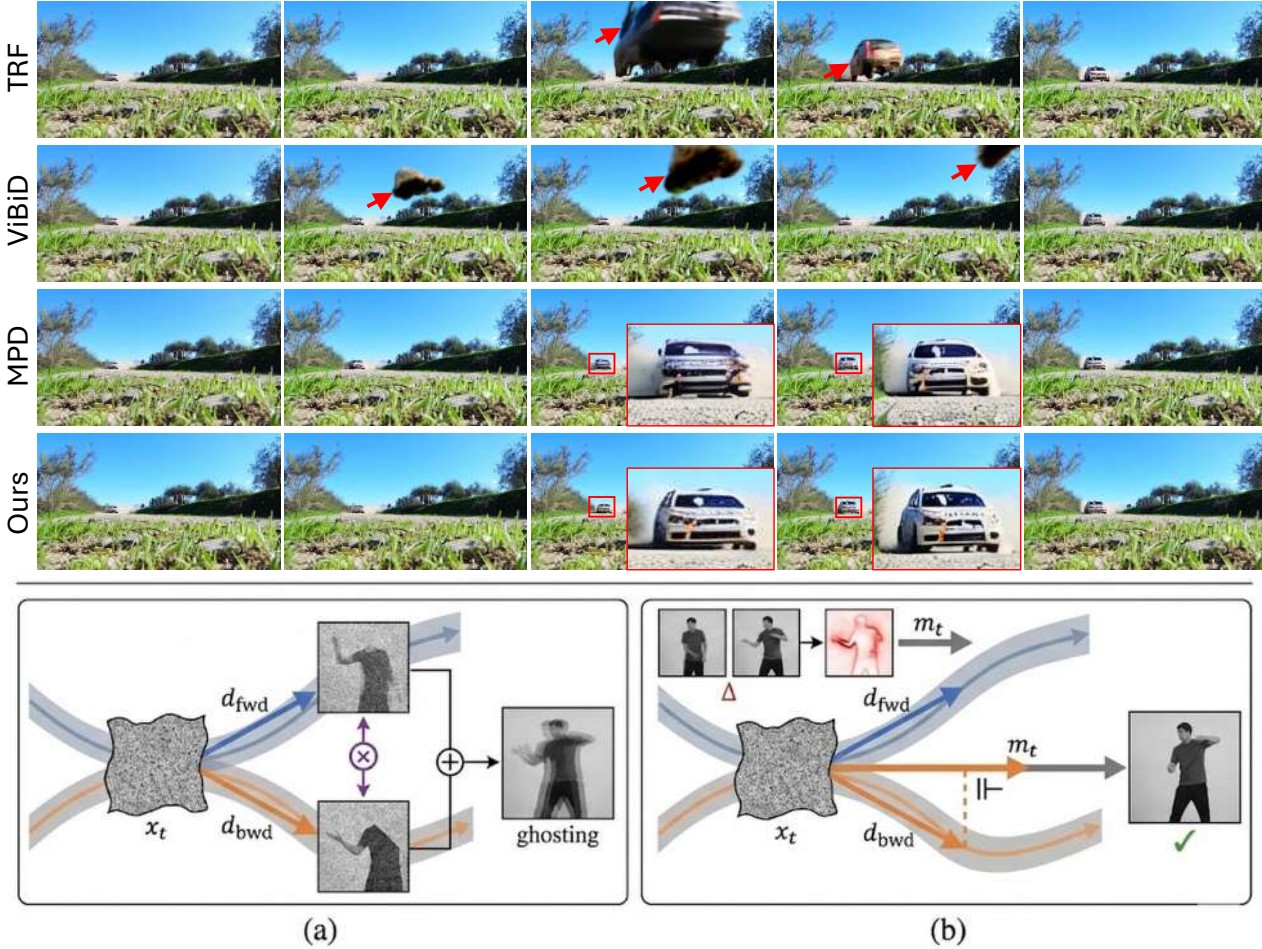

*Figure 1.* **Qualitative comparisons of generative inbetweening results and the core motivation of MR-CATR.** Top: intermediate frame sequences produced by representative time-reversal samplers (TRF/ViBiD/MPD) versus MR-CATR, highlighting typical artifacts such as ghosting and inconsistent motion. Bottom: schematic illustration of local bidirectional motion conflict at a single denoising step and our residual-axis alignment with conflict-aware gated fusion that yields a coherent mid-time state.

conditioned prior can bias the entire trajectory toward an inaccurate explanation of the endpoint.

In this work, we reiterate that the core difficulty lies not only in designing a better fusion schedule or strictly relying on the correctness of a single forward motion prior, but also in aligning the conflicting motion priors induced by the two conditioning frames. We propose Motion-Residual Conflict-Aware Time Reversal (MR-CATR), an inference-time sampling framework that aligns the motion priors of the forward and backward paths without collapsing them into a single start-conditioned prior (Figure 2 (d)). Instead of reconstructing the entire backward denoising trajectory from temporal residuals, MR-CATR uses these residuals to define a shared motion axis in the latent space. As illustrated in Figure 1, at each reverse diffusion step, we first obtain the standard start- and end-conditioned updates, then project the backward update onto this motion axis, and finally perform conflict-aware fusion between the forward update and the

aligned backward update. This design has two major advantages. First, MR-CATR explicitly suppresses bidirectional motion conflicts, reducing ghosting and reverse motion. Second, because we never overwrite the backward trajectory, MR-CATR remains robust even when the forward trajectory is imperfect, and can naturally incorporate motion cues that are observable only near the end frame. In addition, we introduce a shared-region mask that applies strong motion-prior alignment only where the two keyframes depict the same objects, further improving robustness in large-scale scene rearrangements.

Our method operates purely at inference time and can be plugged into existing time reversal samplers such as TRF and ViBiD without modifying model parameters. Through extensive experiments on standard generative inbetweening benchmarks, we show that MR-CATR consistently improves temporal coherence, reduces visual artifacts. User studies further confirm that videos generated by MR-CATR are

preferred in terms of motion plausibility and perceptual quality, especially in scenarios with complex dynamics and large temporal gaps between keyframes.

## 2. Preliminaries

### 2.1. Stable Video Diffusion

Our formulation builds upon Stable Video Diffusion (SVD) (Blattmann et al., 2023), a UNet-based latent video diffusion model widely adopted as the backbone for time-reversal inbetweening methods (Yang et al., 2024; Feng et al., 2024). SVD follows the EDM parameterization (Karras et al., 2022). At a reverse diffusion step $t \in \{T, \ldots, 1\}$ with noise level $\sigma_t$, the denoiser $D_\theta$ predicts both unconditional and conditional clean latents from the noisy latent $x_t$:

$$\hat{x}_{0,\varnothing} = D_\theta(x_t; \sigma_t), \qquad \hat{x}_{0,c} = D_\theta(x_t; \sigma_t, c), \quad (1)$$

where $c$ denotes the conditioning signal.

In EDM, the denoiser is related to the noise prediction and score function as

$$s_\theta(x_t; \sigma_t) = -\frac{\varepsilon_\theta(x_t; \sigma_t)}{\sigma_t} = \frac{D_\theta(x_t; \sigma_t) - x_t}{\sigma_t^2}. \quad (2)$$

Classifier-free guidance (CFG) (Ho & Salimans, 2022) is applied by combining conditional and unconditional predictions:

$$\hat{x}_{0,c} \leftarrow (1 + w)\hat{x}_{0,c} - w\hat{x}_{0,\varnothing}, \qquad w \geq 0. \quad (3)$$

Given $\hat{x}_{0,c}$, a single Euler step updates the sample as

$$x_{t-1} = \hat{x}_{0,c} + \frac{\sigma_{t-1}}{\sigma_t}(x_t - \hat{x}_{0,c}), \quad (4)$$

progressively denoising from $x_T$ to $x_0$.

In standard I2V generation, SVD conditions on a single starting frame. For bounded inbetweening, time-reversal-based methods introduce two diffusion paths, each conditioned on one endpoint and perform reverse diffusion.

### 2.2. Time Reversal Sampling

**Parallel formulation.** Parallel time-reversal methods maintain two diffusion paths conditioned on the start and end frames, respectively, and combine them at each denoising step (Feng et al., 2024; Zhu et al., 2025; Wang et al., 2025). Let $c_{start}$ and $c_{end}$ denote the latent conditions of the start and end frames. A generic parallel update is given by

$$x_{t-1} = \alpha\, x_{t-1,c_{start}} + (1 - \alpha)\,(x'_{t-1,c_{end}})', \qquad \alpha \in [0, 1], \quad (5)$$

where the forward and backward updates follow standard reverse Euler steps,

$$x_{t-1,c_{start}} = \hat{x}_{0,c_{start}} + \frac{\sigma_{t-1}}{\sigma_t}(x_t - \hat{x}_{0,c_{start}}), \quad (6)$$

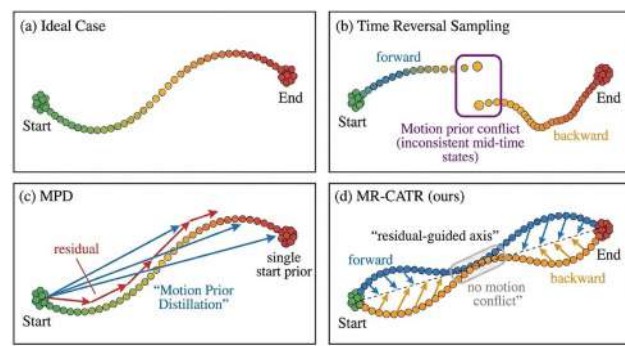

*Figure 2.* (a) Ideal inbetweening follows a single coherent trajectory. (b) Time-reversal sampling induces conflicting start- and end-conditioned priors. (c) MPD resolves this by collapsing both paths into a start-conditioned prior. (d) MR-CATR aligns bidirectional priors along a residual-guided axis, reducing conflict while preserving end-frame motion cues.

$$x'_{t-1,c_{end}} = \hat{x}'_{0,c_{end}} + \frac{\sigma_{t-1}}{\sigma_t}(x'_t - \hat{x}'_{0,c_{end}}), \quad (7)$$

and $(\cdot)'$ denotes temporal reversal.

This linear fusion does not explicitly reconcile the motion priors implied by the two paths. As a result, the blended state may lie between incompatible trajectories, leading to degraded fidelity when the start- and end-conditioned motions disagree.

**Sequential formulation.** An alternative is to denoise the two paths sequentially (Yang et al., 2024) The sampler first follows the start-conditioned path, then injects a small amount of noise to re-randomize the state, and finally continues along the end-conditioned path. Concretely, the forward segment is

$$x_{t-1,c_{start}} = \hat{x}_{0,c_{start}} + \frac{\sigma_{t-1}}{\sigma_t}(x_t - \hat{x}_{0,c_{start}}), \quad (8)$$

followed by a single re-noising step

$$x_{t,c_{start}} = x_{t-1,c_{start}} + \sqrt{\sigma_t^2 - \sigma_{t-1}^2}\,\varepsilon, \qquad \varepsilon \sim \mathcal{N}(0, I), \quad (9)$$

and then an end-conditioned backward update

$$x_{t-1} = \left(\hat{x}_{0,c_{end}} + \frac{\sigma_{t-1}}{\sigma_t}(x_{t,c_{start}} - \hat{x}_{0,c_{end}})\right)'. \quad (10)$$

Compared to parallel schemes, sequential time reversal keeps samples closer to the learned data manifold by following valid denoising trajectories. However, forward and backward segments remain governed by independent start- and end-conditioned motion priors, and switching between them can introduce inconsistent motion directions. As a result, temporal incoherence is not fully resolved. This motivates our MR-CATR, which explicitly aligns the motion priors of the two paths at inference time rather than relying solely on their fusion.

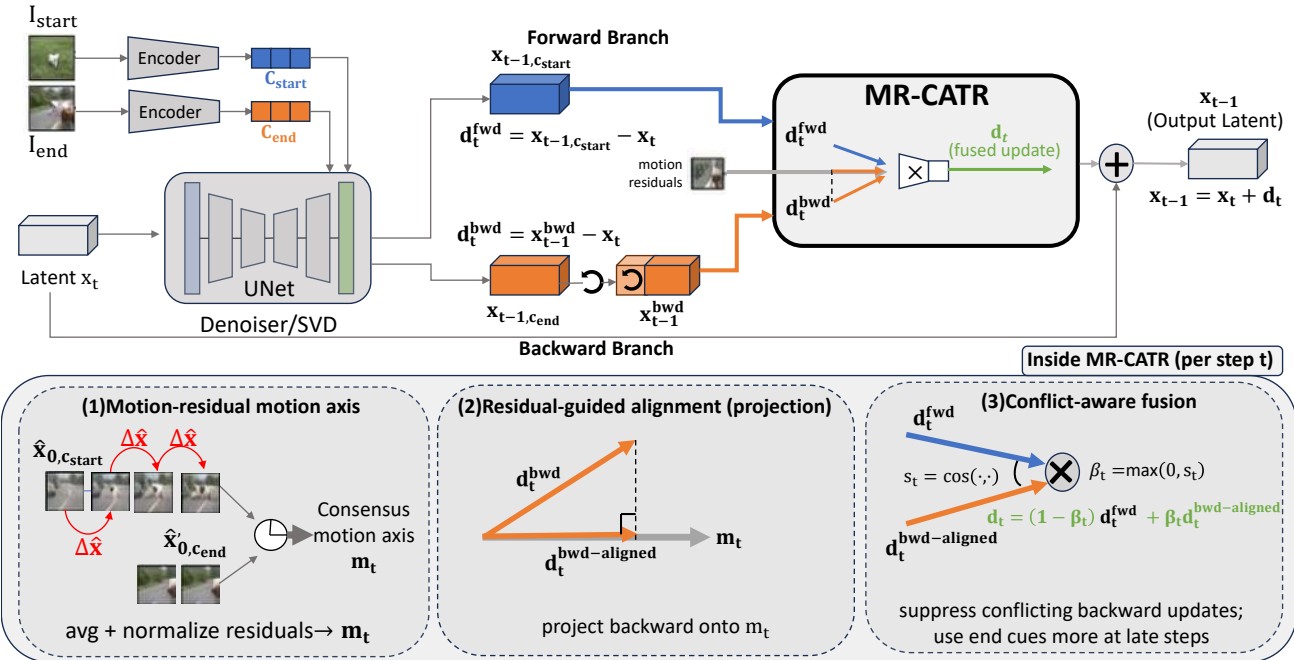

*Figure 3.* **Overview of MR-CATR for time-reversal sampling.** Top: start- and end-conditioned branches produce forward and backward updates from the current latent $x_t$, which are then fused by MR-CATR to obtain $x_{t-1}$. Bottom: (1) motion-residuals define a consensus motion axis $m_t$; (2) the backward update is projected onto $m_t$ for residual-guided alignment; (3) conflict-aware fusion gates the aligned backward update by agreement and a noise-level schedule.

## 3. Method

### 3.1. Motion-Residual Based Motion Prior

At reverse diffusion step $t$, we use $\hat{x}_{0|t,c}$ to denote the clean latent estimate predicted from the current noisy latent $x_t$ under condition $c$. Here, the subscript $0 \mid t$ means an estimate of $x_0$ conditioned on the current step $t$, not a variable at timestep zero. We use $i \in \{1, \ldots, N\}$ to index video frames and $(\cdot)'$ to denote temporal reversal.

Let $\hat{x}_{0|t,c_{\text{start}}} \in \mathbb{R}^{C \times N \times H \times W}$ be the start-conditioned denoised estimate at step $t$, and write $\hat{x}_{0|t,c_{\text{start}}}^{(i)}$ for its $i$-th frame. Following the observation that temporal residuals encode motion information, we define

$$\Delta \hat{x}_{0|t,c_{\text{start}}}^{(i)} = \hat{x}_{0|t,c_{\text{start}}}^{(i)} - \hat{x}_{0|t,c_{\text{start}}}^{(i-1)}, \quad i = 2, \ldots, N. \quad (11)$$

These residuals capture how the model expects the scene to move between neighboring frames under the start-conditioned motion prior.

To obtain a global motion direction at step $t$, we average the residuals over time and normalize:

$$m_t^{\text{start}} = \text{normalize}\left(\frac{1}{N-1}\sum_{i=2}^{N}\Delta \hat{x}_{0|t,c_{\text{start}}}^{(i)}\right), \quad (12)$$

where $\text{normalize}(v) = v/(\|v\|_2 + \varepsilon)$ and $\varepsilon > 0$ avoids division by zero. Intuitively, $m_t^{\text{start}}$ represents the dominant

motion direction implied by the start-conditioned trajectory at the current noise level.

To mitigate the asymmetry of relying only on the start frame, we also compute an end-conditioned motion residual. Let $\hat{x}'_{0|t,c_{\text{end}}}$ denote the end-conditioned denoised estimate in time-reversed order. Here, "time-reversed order" means that the end-conditioned branch is generated with the end frame as the temporal anchor, but its frame index is interpreted after reversing the temporal dimension back to the inbetweening direction. Thus, $\hat{x}'^{(i)}_{0|t,c_{\text{end}}}$ denotes the $i$-th frame of the time-reversed end-conditioned estimate when aligned with the target start-to-end video order. Equivalently, it corresponds to the $i$-th frame along the inbetweening direction, rather than the raw end-to-start order of the end-conditioned branch. We define

$$\Delta \hat{x}'^{(i)}_{0|t,c_{\text{end}}} = \hat{x}'^{(i)}_{0|t,c_{\text{end}}} - \hat{x}'^{(i-1)}_{0|t,c_{\text{end}}}, \quad i = 2, \ldots, N, \quad (13)$$

and its global direction as

$$m_t^{\text{end}} = \text{normalize}\left(\frac{1}{N-1}\sum_{i=2}^{N}\Delta \hat{x}'^{(i)}_{0|t,c_{\text{end}}}\right). \quad (14)$$

We measure the agreement between the two motion priors by

$$s_t^{\text{prior}} = \frac{\langle m_t^{\text{start}}, m_t^{\text{end}}\rangle}{\|m_t^{\text{start}}\|_2 \|m_t^{\text{end}}\|_2 + \varepsilon}, \quad (15)$$

and construct the consensus motion axis

$$m_t = \begin{cases} \text{normalize}\left((1-\kappa)m_t^{\text{start}} + \kappa m_t^{\text{end}}\right), & s_t^{\text{prior}} \geq 0, \\ m_t^{\text{start}}, & s_t^{\text{prior}} < 0. \end{cases}$$
(16)

Here, $\kappa \in [0, 1]$ balances the contributions of the two priors. In our experiments, we use a fixed $\kappa = 0.2$. When the two priors strongly disagree, i.e., $s_t^{\text{prior}} < 0$, we fall back to the start-conditioned motion axis.

### 3.2. Residual-Guided Alignment of Backward Updates

In the parallel TRF sampler, the start- and end-conditioned states at step $t - 1$ are given by Eqs. (6)–(7). We denote the end-conditioned state after temporal reversal as

$$x_{t-1}^{\text{bwd}} = (x'_{t-1,c_{\text{end}}})'.$$
(17)

The corresponding update directions are

$$d_t^{\text{fwd}} = x_{t-1,c_{\text{start}}} - x_t, \qquad d_t^{\text{bwd}} = x_{t-1}^{\text{bwd}} - x_t.$$
(18)

MR-CATR uses the motion axis $m_t$ to align the backward update direction with the residual-based motion prior. Instead of reconstructing a new backward trajectory, we project the backward update onto $m_t$:

$$d_t^{\text{bwd-aligned}} = \langle d_t^{\text{bwd}}, m_t \rangle m_t.$$
(19)

This removes the component of $d_t^{\text{bwd}}$ that is orthogonal to the motion axis, while preserving the magnitude of the end-conditioned update along the shared motion direction. More details are provided in Appendix C.

### 3.3. Conflict-Aware Fusion

After aligning the backward update, MR-CATR decides how much to trust it relative to the forward update. The effective fusion weight depends on both directional agreement and the current noise level. This fusion schedule is independent of the underlying diffusion noise schedule $\{\sigma_t\}_{t=1}^T$, which is inherited unchanged from the base sampler.

**Directional agreement.** For the parallel time-reversal formulation, we compute the cosine similarity between the forward update and the aligned backward update:

$$s_t = \frac{\langle d_t^{\text{fwd}}, d_t^{\text{bwd-aligned}} \rangle}{\|d_t^{\text{fwd}}\|_2 \|d_t^{\text{bwd-aligned}}\|_2 + \varepsilon}.$$
(20)

If $s_t \approx 1$, the two updates are well aligned; if $s_t \leq 0$, they are conflicting or point in opposite directions.

**Noise-level fusion schedule.** We define a monotonic fusion schedule

$$\rho_t = 1 - \left(\frac{\sigma_t}{\sigma_T}\right)^p, \qquad p = 2,$$
(21)

where $\sigma_T$ is the largest noise level in the sampling trajectory. Since $\sigma_t$ decreases during reverse diffusion, $\rho_t$ increases as sampling proceeds. Through the effective fusion weight in Eq. (22), this means that $\beta_t$ is small at early, high-noise steps with large $t$, and can become larger at later, low-noise steps with small $t$, provided that the directional agreement is positive. Thus, within the active MR-CATR steps, aligned backward corrections are suppressed at very noisy steps and receive larger weights at lower noise levels.

**Effective fusion weight.** The final fusion weight is

$$\beta_t = \rho_t \max(0, s_t).$$
(22)

This suppresses backward updates with negative directional agreement and gradually increases the contribution of well-aligned backward updates as the noise level decreases.

The final update direction is

$$d_t = (1 - \beta_t)d_t^{\text{fwd}} + \beta_t d_t^{\text{bwd-aligned}},$$
(23)

and the latent state is updated as

$$x_{t-1} = x_t + d_t.$$
(24)

This replaces the linear interpolation in Eq. (5). When $\beta_t \approx 0$, MR-CATR follows the start-conditioned path; when $\beta_t$ is large and $s_t \approx 1$, the end-conditioned path refines the trajectory along the shared motion axis.

**Sequential sampler.** For the sequential ViBiD instantiation, the forward and backward half-steps are separated by re-noising and therefore do not provide two candidate updates from the same reference state. MR-CATR is instead applied as an axis-aligned correction to the end-conditioned backward half-step after temporal reversal. Let $x_t^{\text{rev}}$ denote the re-noised latent after temporal reversal, which serves as the input to the end-conditioned backward half-step. We denote by $d_t^{\text{seq-bwd}}$ the raw update direction produced by this backward half-step from $x_t^{\text{rev}}$. The motion axis $m_t^{\text{rev}}$ denotes the consensus motion axis expressed in the same reversed temporal order, and $d_t^{\text{seq-bwd-aligned}}$ denotes the projection of $d_t^{\text{seq-bwd}}$ onto $m_t^{\text{rev}}$, analogous to Eq. (34). We use the same noise-level schedule $\rho_t$, but replace the bidirectional agreement score with an axis-agreement score:

$$a_t^{\text{seq}} = \frac{\langle d_t^{\text{seq-bwd}}, m_t^{\text{rev}} \rangle}{\|d_t^{\text{seq-bwd}}\|_2 \|m_t^{\text{rev}}\|_2 + \varepsilon},$$
(25)

and set

$$\beta_t^{\text{seq}} = \rho_t \max(0, a_t^{\text{seq}}).$$
(26)

The corrected backward half-step is

$$x_{t-1}^{\text{rev}} = x_t^{\text{rev}} + (1 - \beta_t^{\text{seq}})d_t^{\text{seq-bwd}} + \beta_t^{\text{seq}} d_t^{\text{seq-bwd-aligned}},$$
(27)

followed by temporal reversal back to the original order. This distinction reflects that the sequential sampler exposes a single backward update after re-noising, rather than parallel forward and backward candidates from the same latent state.

### 3.4. Shared-Region Mask

Time-reversal sampling is most reliable in regions where the two keyframes depict the same objects undergoing semantically coherent motion. In contrast, when the end keyframe introduces new objects or large-scale scene rearrangements, enforcing strong alignment to the start-conditioned motion prior can be harmful. To improve robustness, we introduce a shared-region mask.

Let $p$ index spatial locations. We construct a binary mask $M(p)$ indicating whether location $p$ is likely to correspond to a shared object or background in both keyframes:

$$M(p) = \begin{cases} 1, & \text{if } p \text{ is in a shared region,} \\ 0, & \text{otherwise.} \end{cases} \quad (28)$$

In practice, $M(p)$ can be obtained by comparing feature embeddings of the start and end frames and thresholding their similarity; here we treat it as given.

We apply residual-guided alignment only in shared regions and leave the raw backward update unchanged elsewhere:

$$d_t^{\text{bwd-mask}}(p) = M(p)d_t^{\text{bwd-aligned}}(p) + (1 - M(p))d_t^{\text{bwd}}(p). \quad (29)$$

The fused update becomes

$$d_t(p) = (1 - \beta_t)d_t^{\text{fwd}}(p) + \beta_t d_t^{\text{bwd-mask}}(p), \quad (30)$$

followed by $x_{t-1}(p) = x_t(p) + d_t(p)$ as in Eq. (24). Thus, MR-CATR applies strong motion-prior alignment where shared-object correspondence is reliable, while preserving flexibility in newly appearing or rearranged regions.

Algorithms 1 and 2 in Appendix describe the sampler-specific pseudocode of MR-CATR.

## 4. Experiments

### 4.1. Experimental Settings

**Baselines and metrics.** We compare our method with a representative set of baselines that span both generative and conventional video frame interpolation approaches. Specifically, we include recent generative inbetweening methods, namely TRF (Feng et al., 2024), GI (Wang et al., 2024b), and ViBiD (Yang et al., 2024). In addition, we consider FILM (Reda et al., 2022), a widely used flow-based frame interpolation model, as a strong non-generative baseline. To reflect recent advances in generative inbetweening, we further include MPD (Jeon et al., 2026), which represents a state-of-the-art approach based on motion-prior-driven time-reversal sampling.

For quantitative evaluation, we report commonly adopted metrics for video generation and interpolation, including FID (Heusel et al., 2017), FVD (Unterthiner et al., 2019),

and LPIPS (Zhang et al., 2018). FID and FVD measure distribution-level discrepancies between generated outputs and ground-truth videos, while LPIPS evaluates perceptual similarity at the frame level. We additionally assess overall video quality using VBench (Huang et al., 2024a), a multi-dimensional benchmark that evaluates aspects such as subject consistency, background consistency, aesthetic quality, and motion smoothness. More details about experimental settings are provided in Appendix E.1.

### 4.2. Comparative Results

**Quantitative results.** Quantitative comparisons on the DAVIS and Pexels datasets are presented in Tables 1 and 2, respectively. As shown in the results, our method consistently achieves superior performance over existing generative video inbetweening approaches across most evaluation metrics, demonstrating its effectiveness under different data distributions and motion characteristics. In particular, when integrated as a plug-and-play module into representative generative inbetweening frameworks such as TRF and ViBiD, the proposed MR-CATR brings clear and consistent improvements in both FID and FVD. These gains verify that our method can effectively complement existing generation pipelines without requiring architectural redesign or additional task-specific retraining. Our method performs motion reasoning through residual-guided alignment and conflict-aware fusion. This design enables MR-CATR to exploit motion cues from both boundary frames more flexibly and to adaptively resolve inconsistent or ambiguous motion patterns. As a result, our approach provides a more robust mechanism for modeling complex temporal transitions.

Furthermore, the VBench results in Table 2 provide a more fine-grained evaluation of the generated video quality. Our method achieves higher scores on most dimensions, including subject consistency, background consistency, aesthetic quality, and motion smoothness, compared with existing generative baselines. These improvements indicate that MR-CATR not only enhances frame-level visual quality but also contributes to more stable object appearance, more coherent background preservation, and smoother motion evolution across time. Overall, by reducing motion inconsistencies and improving the reliability of bidirectional motion integration, MR-CATR substantially improves the overall quality and plausibility of generated inbetweening videos.

**Qualitative results.** Figure 4 presents qualitative comparisons between MR-CATR and representative generative inbetweening methods based on the time-reversal paradigm. We observe that TRF and ViBiD exhibit noticeable misalignment near both the starting and ending keyframes, reflecting the inherent path inconsistency between the two generation directions (e.g., the disappearance of the bus stop sign in Case 1, indicated by the red arrow). Such misalignment

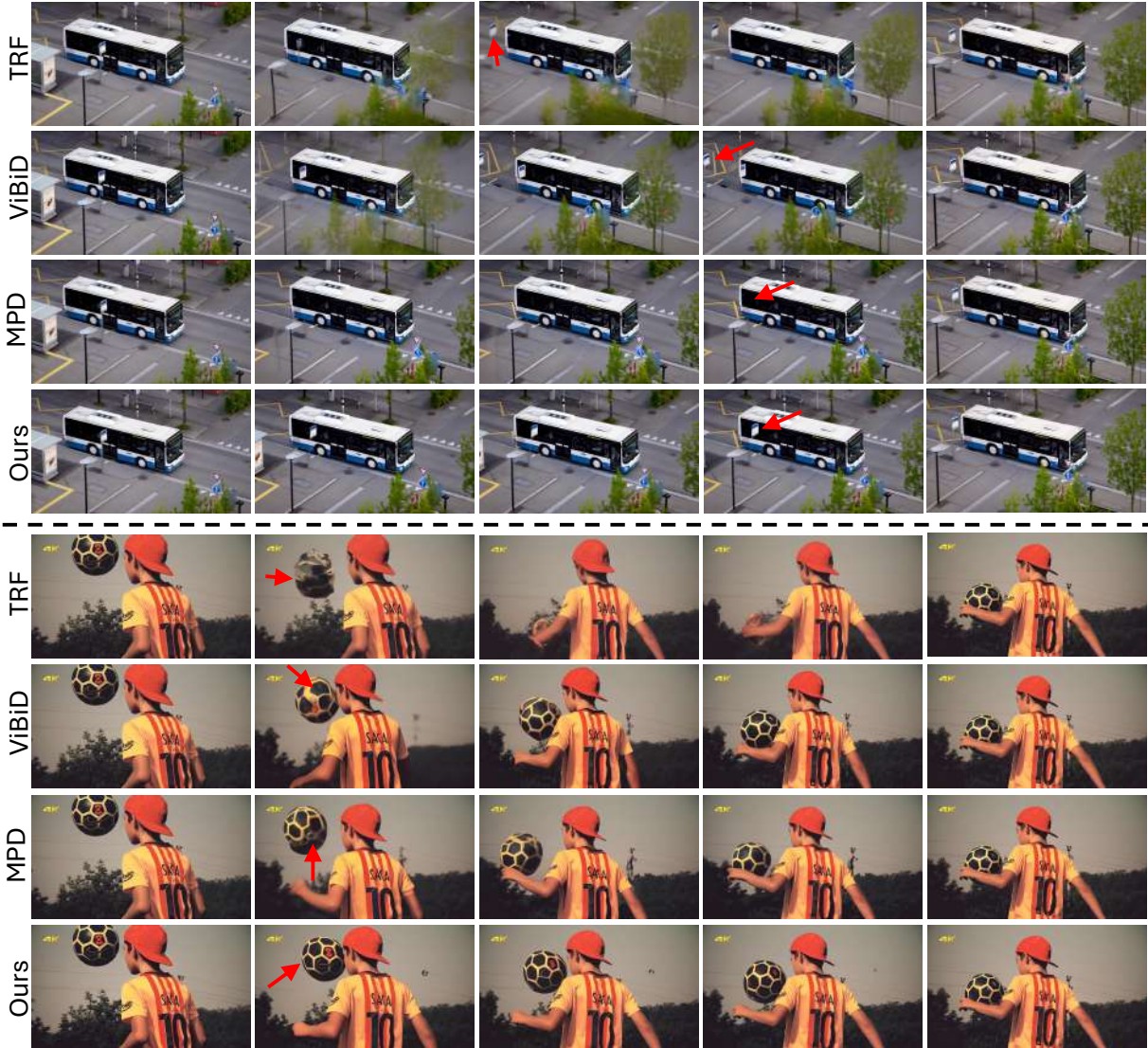

*Figure 4.* **Qualitative comparisons of generative results.** Our method exhibits improved temporal consistency compared to the baselines.

often leads to unnatural motion trajectories and blurry artifacts in the synthesized intermediate frames. MPD injects trajectory residuals conditioned on the starting frame while ignoring the informational constraints from the ending frame, which leads to inconsistent object appearance (e.g., surface artifacts on the soccer ball in Case 2). In contrast, our method explicitly integrates conditions from both the start and end frames, enabling more coherent trajectory alignment throughout the sequence. As a result, MR-CATR produces intermediate frames with smoother and more temporally consistent motion, clearer object boundaries, and more stable motion trajectories. Additional examples are provided in Appendix F.3.

**Human evaluation.** Since existing automatic metrics do not fully capture perceptual realism and temporal coherence

in generative inbetweening, we further conduct a human evaluation study. The study focuses on perceptual aspects that are difficult to measure automatically, including overall user preference, visual artifact severity, and motion plausibility. Additional implementation details are provided in Appendix D. Figure 5 summarizes the results of the human evaluation. Our method consistently achieves higher user preference while exhibiting lower rates of visual artifacts and implausible motion compared to all baseline methods.

### 4.3. Ablation Studies

**Effect of motion axis construction.** We investigate the effect of constructing the consensus motion axis by incorporating both start- and end-conditioned residuals. To isolate this factor, we compare the full model with a variant

| Method | DAVIS | | | Pexels | | |
|---|---|---|---|---|---|---|
| | LPIPS↓ | FID↓ | FVD↓ | LPIPS↓ | FID↓ | FVD↓ |
| FILM (Reda et al., 2022) | 0.2843 | 47.124 | 942.55 | 0.2434 | 45.397 | 732.10 |
| TRF (Feng et al., 2024) | 0.3241 | 53.626 | 643.92 | 0.2479 | 52.408 | 608.54 |
| GI (Wang et al., 2024b) | 0.2465 | 46.807 | 612.89 | 0.2527 | 41.843 | 543.77 |
| ViBiD (Yang et al., 2024) | 0.2406 | 41.586 | 548.25 | 0.2481 | 39.577 | 517.84 |
| MPD+TRF (Jeon et al., 2026) | 0.2675 | 36.720 | 528.03 | 0.1971 | 35.520 | 489.02 |
| MPD+ViBiD (Jeon et al., 2026) | 0.2316 | 38.157 | 527.44 | 0.1756 | 36.189 | 488.45 |
| Ours+TRF | 0.2286 | **35.482** | 523.56 | 0.1656 | **35.261** | 467.23 |
| Ours+ViBiD | **0.2103** | 36.930 | **519.47** | **0.1598** | 37.062 | **462.77** |

*Table 1.* Quantitative comparison on DAVIS and Pexels datasets.

| Method | DAVIS | | | | Pexels | | | |
|---|---|---|---|---|---|---|---|---|
| | Subj.↑ | Motion↑ | Bg.↑ | Aesth.↑ | Subj.↑ | Motion↑ | Bg.↑ | Aesth.↑ |
| TRF (Feng et al., 2024) | 0.8825 | 0.9267 | 0.8912 | 0.6623 | 0.8547 | 0.9032 | 0.8619 | 0.6345 |
| GI (Wang et al., 2024b) | 0.9014 | 0.9481 | 0.9135 | 0.6718 | 0.8713 | 0.9278 | 0.8796 | 0.6418 |
| ViBiD (Yang et al., 2024) | 0.9156 | 0.9542 | 0.9217 | 0.6784 | 0.8849 | 0.9361 | 0.8924 | 0.6427 |
| MPD+TRF (Jeon et al., 2026) | 0.9238 | 0.9627 | 0.9289 | 0.6841 | 0.8962 | 0.9489 | 0.9047 | 0.6553 |
| MPD+ViBiD (Jeon et al., 2026) | 0.9289 | 0.9678 | 0.9315 | 0.6875 | 0.9021 | **0.9526** | 0.9098 | **0.6581** |
| Ours+TRF | 0.9327 | **0.9735** | **0.9432** | **0.6896** | 0.9138 | 0.9484 | 0.9156 | 0.6472 |
| Ours+ViBiD | **0.9520** | 0.9643 | 0.9226 | 0.6743 | **0.9274** | 0.9511 | **0.9238** | 0.6424 |

*Table 2.* VBench evaluation of generative inbetweening on DAVIS and Pexels datasets.

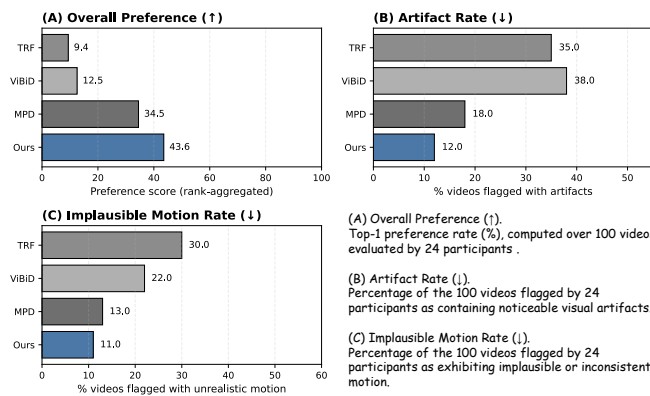

(A) Overall Preference (↑).
Top-1 preference rate (%), computed over 100 videos evaluated by 24 participants.

(B) Artifact Rate (↓).
Percentage of the 100 videos flagged by 24 participants as containing noticeable visual artifacts.

(C) Implausible Motion Rate (↓).
Percentage of the 100 videos flagged by 24 participants as exhibiting implausible or inconsistent motion.

*Figure 5.* Human evaluation results across preference, artifact, and motion plausibility.

| Method | Variant | LPIPS↓ | FID↓ | FVD↓ |
|---|---|---|---|---|
| Ours+TRF | w/o End-Res | 0.236 | 38.9 | 548.3 |
| | Full | **0.228** | **35.5** | **523.6** |
| Ours+ViBiD | w/o End-Res | 0.221 | 37.8 | 531.4 |
| | Full | **0.210** | **36.9** | **519.5** |

*Table 3.* Ablation results on motion axis construction. Incorporating end-conditioned residuals consistently improves temporal consistency, with the most significant gains observed in FVD.

that builds the motion axis solely from start-conditioned denoised estimates. As reported in Table 3, the start-only variant performs reasonably on simple sequences where the start-conditioned trajectory already provides a reliable motion prior. However, its performance deteriorates when important motion cues are mainly revealed near the ending keyframe or when the motion trajectory must be adjusted to satisfy the terminal condition. This degradation is most pronounced in terms of FVD, indicating reduced temporal consistency and increased motion-related artifacts.

The qualitative results in Figure 6 further support this observation. Without the terminal-condition residual, the generated trajectory tends to drift away from the desired endpoint in the later part of the sequence. This often leads to incorrect human structures, distorted poses, and unstable motion around Frames 20–22. In contrast, the full model uses the end-conditioned residual to refine the consensus motion axis, allowing the trajectory to progressively converge toward the final frame. As a result, the generated sequence exhibits smoother motion, fewer late-stage artifacts, and better temporal coherence throughout the video.

**Effect of residual-guided projection.** We study the role of the residual-guided projection by removing the projection step and directly applying conflict-aware gating to the

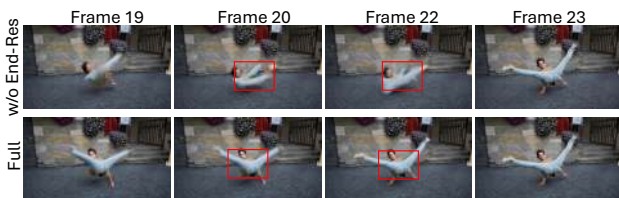

*Figure 6.* Qualitative comparison of motion axis construction with and without end-conditioned residuals.

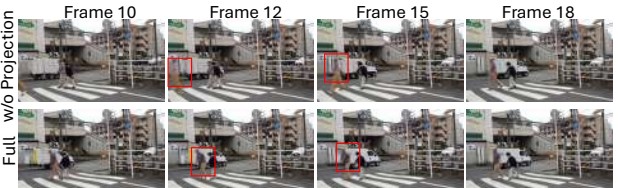

*Figure 7.* Qualitative comparison of backward updates with and without projection. Removing the projection step leads to lateral drift and minor temporal jitter.

| Method | Variant | LPIPS↓ | FID↓ | FVD↓ |
|---|---|---|---|---|
| Ours+TRF | w/o projection | 0.241 | 39.2 | 552.2 |
| | Full | **0.228** | **35.5** | **523.6** |
| Ours+ViBiD | w/o projection | 0.247 | 39.1 | 547.4 |
| | Full | **0.210** | 36.9 | **519.5** |

*Table 4.* Ablation results on residual-guided projection. Removing the projection step degrades temporal consistency compared to the full model.

| Active window | MR-CATR on TRF | | MR-CATR on ViBiD | |
|---|---|---|---|---|
| | FVD↓ | LPIPS↓ | FVD↓ | LPIPS↓ |
| Early $0.2T$ | **467.2** | **0.166** | **462.8** | **0.160** |
| Late $0.2T$ | 589.6 | 0.222 | 578.4 | 0.215 |
| Early $0.5T$ | 471.9 | 0.167 | 466.1 | 0.161 |
| All $T$ | 476.5 | 0.169 | 470.6 | 0.162 |

*Table 5.* Ablation on the active alignment window of MR-CATR.

backward update. As shown in Table 4, while this variant outperforms naive time-reversal sampling baselines, it consistently underperforms the full MR-CATR model. Qualitative results reveal residual lateral drift and temporal jitter, particularly in cases where the backward update contains components orthogonal to the dominant motion direction. As illustrated in Figure 7, removing the projection step leads to noticeable artifacts in human motion, such as ghosting and inconsistent limb trajectories, whereas the full model produces smoother and more coherent pedestrian motion. These observations indicate that the residual-guided projection plays a critical role in constraining backward updates to a coherent motion subspace, thereby improving motion consistency.

**Effect of active alignment window.** We further study when MR-CATR should be activated during the reverse diffusion process. Since sampling proceeds from $t = T$ to $t = 1$, the "early" steps refer to the highest-noise stage. In our default setting, MR-CATR is applied only to the first $0.2T$ reverse steps:

$$\mathcal{T}_{\text{align}} = \{T, T-1, \ldots, T - \lfloor 0.2T \rfloor + 1\}. \quad (31)$$

For $T = 25$, this corresponds to the first 5 denoising steps.

At high noise levels, the sampler mainly determines the global structure of the generated video, including the rough object layout and the dominant motion trajectory. In this stage, inconsistent start- and end-conditioned motion priors can easily lead to an incorrect global path, which then propagates to later denoising steps. Therefore, applying MR-CATR early helps suppress trajectory-level conflict before the video structure becomes fixed.

In contrast, later low-noise steps are more closely related

to appearance, texture, and endpoint-detail refinement. At this stage, the coarse motion layout has largely been established, and applying the same residual-guided alignment too aggressively may over-constrain local details or reduce the flexibility needed to match the final keyframe. Thus, we activate MR-CATR only in the early high-noise window, where motion-prior alignment is most useful and least likely to interfere with fine-grained visual refinement.

As shown in Table 5, activating MR-CATR during the early $0.2T$ highest-noise steps gives the best performance for both TRF and ViBiD. Applying the alignment only in the late $0.2T$ lowest-noise steps performs noticeably worse, suggesting that late intervention mainly perturbs detail refinement rather than resolving trajectory-level conflict. Extending the active window to early $0.5T$ or all steps also brings no further gain. These results indicate that MR-CATR is most effective as an early-stage motion-prior alignment module. More ablation experiments are provided in Appendix E.2.

## 5. Conclusion

In this work, we propose MR-CATR, an inference-time sampling framework that aligns forward and backward updates through residual-guided motion axes and conflict-aware fusion. MR-CATR effectively suppresses ghosting and reverse-motion artifacts while preserving endpoint consistency. Extensive experiments on DAVIS and Pexels demonstrate that MR-CATR consistently improves temporal coherence and perceptual quality over state-of-the-art baselines. We believe that the proposed residual-guided alignment principle can be extended to other bidirectional or constraint-driven generative tasks and serves as a step toward more coherent and controllable video generation.

## Impact Statement

This paper presents work whose goal is to advance the field of machine learning. While such technologies may have broader societal implications, including both beneficial and potentially harmful applications, we do not identify any unique ethical concerns arising from this work.

## Acknowledgements

This research was supported by the National Key Research and Development Program of China [2021YFF0704300], Dubai Future Foundation (Award No. 2024CANAD-MES-061), and the Natural Science Foundation of Shandong Province [ZR2023YQ057].

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

---

**Algorithm 1** TRF+MR-CATR

---

**Require:** Noise schedule $\{\sigma_t\}_{t=1}^T$; conditions $c_{\text{start}}, c_{\text{end}}$; guidance scale $w$; fusion schedule $\rho_t$; active steps $\mathcal{T}_{\text{align}}$; TRF ratio $\alpha$; mask $M(p)$.

**Ensure:** Decoded video from $x_0$.

  1: Initialize $x_T \sim \mathcal{N}(0, \sigma_T^2 I)$.
  2: **for** $t = T, T-1, \ldots, 1$ **do**
  3:     *1. Bidirectional Denoising*
  4:     Compute start-conditioned candidate $x_{t-1,c_{\text{start}}}$ from $x_t$ via Eq. (6).
  5:     Compute end-conditioned candidate $x'_{t-1,c_{\text{end}}}$ from $x'_t = (x_t)'$ via Eq. (7).
  6:     Set $x_{t-1}^{\text{bwd}} \leftarrow (x'_{t-1,c_{\text{end}}})'$ and $\hat{x}_{0,c_{\text{end}}}^{\text{bwd}} \leftarrow (\hat{x}'_{0,c_{\text{end}}})'$.
  7:     **if** $t \in \mathcal{T}_{\text{align}}$ **then**
  8:         *2. Motion Axis Construction*
  9:         Compute $m_t^{\text{start}}$ from $\hat{x}_{0,c_{\text{start}}}$ and $m_t^{\text{end}}$ from $\hat{x}_{0,c_{\text{end}}}^{\text{bwd}}$ via Eqs. (11)–(14).
10:         Obtain consensus motion axis $m_t$ via Eqs. (15)–(16).
11:         *3. Residual-Guided Alignment*
12:         Compute $d_t^{\text{fwd}} = x_{t-1,c_{\text{start}}} - x_t$ and $d_t^{\text{bwd}} = x_{t-1}^{\text{bwd}} - x_t$.
13:         Project backward update: $d_t^{\text{bwd-aligned}} \leftarrow \langle d_t^{\text{bwd}}, m_t \rangle m_t$.
14:         Apply mask if available: $d_t^{\text{bwd-mask}} \leftarrow M d_t^{\text{bwd-aligned}} + (1 - M) d_t^{\text{bwd}}$; otherwise $d_t^{\text{bwd-mask}} \leftarrow d_t^{\text{bwd-aligned}}$.
15:         *4. Conflict-Aware Fusion*
16:         Compute $s_t$ via Eq. (20) and $\beta_t \leftarrow \rho_t \max(0, s_t)$.
17:         Update $x_{t-1} \leftarrow x_t + (1 - \beta_t) d_t^{\text{fwd}} + \beta_t d_t^{\text{bwd-mask}}$.
18:     **else**
19:         Use vanilla TRF fusion: $x_{t-1} \leftarrow \alpha x_{t-1,c_{\text{start}}} + (1 - \alpha) x_{t-1}^{\text{bwd}}$.
20:     **end if**
21: **end for**

---

# A. Related Work

**Video Frame Interpolation.** Video frame interpolation (VFI) focuses on synthesizing intermediate frames between two given frames while preserving both spatial structure and temporal consistency (Kye et al., 2025; Kiefhaber et al., 2024). A large body of supervised VFI methods (Zhang et al., 2024; Lu et al., 2022; Zhang et al., 2023; Liu et al., 2024; Guo et al., 2024) relies on explicit optical flow estimation followed by warping. These approaches are widely used in practice because they offer robust performance and interpretable motion trajectories. However, inaccuracies in the estimated flow easily lead to artifacts, especially in the presence of occlusions, complex non-rigid motion, or strongly non-linear trajectories (Long et al., 2024). More recently, diffusion-based VFI models (Danier et al., 2024; Huang et al., 2024b) have been proposed to exploit the generative power of diffusion models and improve the perceptual realism of interpolated frames. Although these methods produce visually pleasing results, their performance still degrades when the temporal gap between two frames becomes large and long-range motion must be inferred rather than locally interpolated.

**Generative Video Inbetweening.** With the development of video diffusion models (Blattmann et al., 2023; Huang et al., 2024a; Shi et al., 2024; Wang et al., 2025), the problem setting has evolved from deterministic VFI to generative inbetweening, which aims to model the set of semantically plausible interpolations between two keyframes. One family of approaches (Jain et al., 2024; Xing et al., 2024; Wang et al., 2024a; Zhang et al., 2025) directly trains diffusion models conditioned on both start and end frames, achieving strong robustness under ambiguous or large motions where traditional VFI often fails. The downside is that such methods require substantial training cost and dedicated architectures. Another line of work instead leverages large pre-trained I2V diffusion models and focuses on sampling strategies. TRF (Feng et al., 2024) introduces a parallel time-reversal scheme that denoises start- and end-conditioned paths at each step and fuses them by linear interpolation. GI (Wang et al., 2025) improves reverse motion quality by fine-tuning temporal self-attention maps to better generate time-reversed sequences. FCVG (Zhu et al., 2025) augments the model with line-based correspondence cues between the two keyframes to reduce ambiguities in the inbetweening path. ViBiDSampler (Yang et al., 2024) instead adopts a sequential time-reversal strategy with an intermediate re-noising step to keep samples closer to the data manifold. Recent Motion Prior Distillation (MPD) (Jeon et al., 2026) collapses the two priors into a unified start-conditioned motion prior, distilling forward motion residuals into the backward path. In contrast, our MR-CATR framework seeks to align rather

---

**Algorithm 2** ViBiD+MR-CATR

---

**Require:** Noise schedule $\{\sigma_t\}_{t=1}^{T}$; conditions $c_{\text{start}}, c_{\text{end}}$; guidance scale $w$; fusion schedule $\rho_t$; active steps $\mathcal{T}_{\text{align}}$; mask $M(p)$.

**Ensure:** Decoded video from $x_0$.

1: Initialize $x_T \sim \mathcal{N}(0, \sigma_T^2 I)$.
2: **for** $t = T, T-1, \ldots, 1$ **do**
3:     *1. Forward Half-Step*
4:     Compute guided start prediction $\bar{x}_{0,c_{\text{start}}}$ using the ViBiD guidance rule.
5:     Compute $x_{t-1,c_{\text{start}}}$ by the ViBiD forward half-step.
6:     Re-noise: $x_{t,c_{\text{start}}} \leftarrow x_{t-1,c_{\text{start}}} + \sqrt{\sigma_t^2 - \sigma_{t-1}^2}\,\varepsilon$.
7:     Reverse temporal order: $x_t^{\text{rev}} \leftarrow (x_{t,c_{\text{start}}})'$.
8:     *2. Backward Half-Step*
9:     Compute guided end prediction $\bar{x}'_{0,c_{\text{end}}}$ from $x_t^{\text{rev}}$ using the ViBiD guidance rule.
10:    Compute raw backward candidate $x_{t-1}^{\text{raw-rev}}$ by the ViBiD backward half-step.
11:    **if** $t \in \mathcal{T}_{\text{align}}$ **then**
12:      *3. Motion Axis Construction*
13:      Compute reversed start prior from $(\bar{x}_{0,c_{\text{start}}})'$ and end prior from $\bar{x}'_{0,c_{\text{end}}}$.
14:      Obtain consensus motion axis $m_t^{\text{rev}}$ via Eqs. (15)–(16).
15:      *4. Backward Correction*
16:      Compute $d_t^{\text{seq-bwd}} = x_{t-1}^{\text{raw-rev}} - x_t^{\text{rev}}$.
17:      Project update: $d_t^{\text{seq-aligned}} \leftarrow \langle d_t^{\text{seq-bwd}}, m_t^{\text{rev}}\rangle m_t^{\text{rev}}$.
18:      Apply mask if available: $d_t^{\text{seq-mask}} \leftarrow M d_t^{\text{seq-aligned}} + (1 - M)d_t^{\text{seq-bwd}}$; otherwise $d_t^{\text{seq-mask}} \leftarrow d_t^{\text{seq-aligned}}$.
19:      Compute axis gate $a_t^{\text{seq}} = \frac{\langle d_t^{\text{seq-bwd}}, m_t^{\text{rev}}\rangle}{\|d_t^{\text{seq-bwd}}\|_2 \|m_t^{\text{rev}}\|_2 + \varepsilon}$ and $\beta_t^{\text{seq}} \leftarrow \rho_t \max(0, a_t^{\text{seq}})$.
20:      Correct $x_{t-1}^{\text{rev}} \leftarrow x_t^{\text{rev}} + (1 - \beta_t^{\text{seq}})d_t^{\text{seq-bwd}} + \beta_t^{\text{seq}}d_t^{\text{seq-mask}}$.
21:    **else**
22:      Use vanilla ViBiD backward half-step: $x_{t-1}^{\text{rev}} \leftarrow x_{t-1}^{\text{raw-rev}}$.
23:    **end if**
24:    Restore temporal order: $x_{t-1} \leftarrow (x_{t-1}^{\text{rev}})'$.
25: **end for**

---

than collapse motion priors: we use motion residuals to define a shared motion axis and perform conflict-aware fusion of start- and end-conditioned updates, suppressing motion conflicts while still preserving complementary information from both endpoints.

| Method | TRF | ViBiD | MPD+TRF | MPD+ViBiD | Ours+TRF | Ours+ViBiD |
|---|---|---|---|---|---|---|
| Runtime (s) | 102 | 117 | 154 | 156 | 161 | 167 |
| Relative Overhead | 1.00× (w.r.t. TRF) | 1.00× (w.r.t. ViBiD) | 1.51× (w.r.t. TRF) | 1.33× (w.r.t. ViBiD) | **1.05×** (w.r.t. MPD+TRF) | **1.07×** (w.r.t. MPF+ViBiD) |

*Table 6.* Runtime comparison. Relative overhead is computed with respect to the corresponding backbone method.

## B. Runtime Analysis

We compare the computational cost of our method with representative inference-guided I2V diffusion approaches based on the time-reversal paradigm, including TRF, ViBiD, and MPD. These methods share the same inference-guided sampling framework as our MR-CATR, ensuring a fair and meaningful comparison. All runtime evaluations are conducted on videos of resolution $1024 \times 576$ with 25 frames. Table 6 summarizes the runtime comparison across different methods. While our method introduces additional computation, the resulting overhead is modest. Specifically, when applied on top of TRF and ViBiD, MR-CATR increases the runtime by only 5–7% compared to MPD. These results indicate that the proposed components incur limited computational overhead while providing improved motion consistency and robustness.

# C. Justification of Residual-Guided Projection

In this section, we provide a detailed justification for the residual-guided projection used in Eq. (34), and explain why projecting the backward update onto the motion axis is valid.

**Common Latent Vector Space.** We begin by clarifying that both the backward update and the motion axis reside in the same latent vector space. In diffusion-based video generation, all latent states produced during sampling, including $x_t$, $x_{t-1,c_{\text{start}}}$, $x'_{t-1,c_{\text{end}}}$, and the denoised estimates $\hat{x}^{(i)}_{0,c}$, are elements of the model's latent space with shape $\mathbb{R}^{C \times N \times H \times W}$. This latent space can be equivalently viewed as a finite-dimensional real vector space by flattening tensors into vectors.

By definition, the backward update at step $t$ is given by

$$d_t^{\text{bwd}} = x_{t-1}^{\text{bwd}} - x_t, \tag{32}$$

which is the difference of two latent vectors and therefore also lies in the same latent space. Similarly, the motion axis $m_t$ is constructed through linear operations on denoised latent estimates, including frame-wise differencing, averaging, and normalization. Since this latent space is closed under addition, subtraction, and scalar multiplication, both $d_t^{\text{bwd}}$ and $m_t$ are guaranteed to be elements of the same real inner-product space. As a result, standard notions of inner products, angles, and orthogonal projections are well defined.

**Residual-Guided Projection via Orthogonal Decomposition.** Given a unit motion axis $m_t$, the backward update $d_t^{\text{bwd}}$ admits a unique orthogonal decomposition,

$$d_t^{\text{bwd}} = \underbrace{\langle d_t^{\text{bwd}}, m_t \rangle m_t}_{d_t^{\parallel}} + \underbrace{\left( d_t^{\text{bwd}} - \langle d_t^{\text{bwd}}, m_t \rangle m_t \right)}_{d_t^{\perp}}, \tag{33}$$

where $d_t^{\parallel} \in \text{span}(m_t)$ is the component aligned with the motion axis and $d_t^{\perp} \perp m_t$ captures orthogonal directions.

We define the aligned backward update by retaining only the motion-consistent component,

$$d_t^{\text{bwd-aligned}} = \langle d_t^{\text{bwd}}, m_t \rangle m_t. \tag{34}$$

From an optimization perspective, this update is the solution to a constrained least-squares problem,

$$d_t^{\text{bwd-aligned}} = \arg \min_{u \in \text{span}(m_t)} \| u - d_t^{\text{bwd}} \|_2^2, \tag{35}$$

whose unique minimizer is the orthogonal projection in Eq. (34), since $\text{span}(m_t)$ is a one-dimensional subspace and $m_t$ has unit norm. Thus, the projection is not heuristic but the optimal $\ell_2$-preserving correction that restricts the backward update to the motion-consistent subspace.

Importantly, the scalar coefficient $\langle d_t^{\text{bwd}}, m_t \rangle$ is preserved, allowing the backward path to encode how strongly the end-conditioned sampler supports motion along the shared axis. Endpoint-related information is therefore retained through the magnitude of the update, while potentially conflicting directional components are suppressed.

This effect can also be interpreted in terms of energy reduction. Projecting $d_t^{\text{bwd}}$ onto $\text{span}(m_t)$ removes the orthogonal component $d_t^{\perp}$, yielding

$$\| d_t^{\text{bwd}} - d_t^{\text{bwd-aligned}} \|_2^2 = \| d_t^{\perp} \|_2^2, \qquad \| d_t^{\text{bwd-aligned}} \|_2^2 = \| d_t^{\text{bwd}} \|_2^2 - \| d_t^{\perp} \|_2^2. \tag{36}$$

Therefore, residual-guided projection provably eliminates the portion of the backward update that is inconsistent with the residual-induced motion axis, leading to more stable and temporally coherent updates.

# D. Human Evaluation Details

This appendix provides a detailed description of the human evaluation protocol used in our experiments.

### D.1. Participants and Evaluation Objectives

The study involves 24 participants. Each participant evaluates a subset of samples, resulting in multiple independent evaluations per video. The goal of the human study is to assess perceptual aspects of generative video inbetweening that are not fully captured by automatic metrics. In particular, we focus on four criteria that are critical for inbetweening quality: (1) Temporal coherence, reflecting smooth and consistent motion progression without jitter or abrupt reversals; (2) Motion plausibility, measuring whether the synthesized motion follows physically and semantically reasonable trajectories; (3) Artifact severity, including visual artifacts such as ghosting, flickering, or deformation; and (4) Endpoint consistency, evaluating whether the generated video remains well aligned with both the starting and ending keyframes.

### D.2. Methods and Data Selection

We evaluate representative generative inbetweening methods, including TRF, ViBiD, MPD, and our proposed method MR-CATR. All methods generate videos conditioned on the same keyframe pairs. Video samples used in the human evaluation are drawn from the DAVIS and Pexels datasets.

### D.3. Evaluation Tasks

Participants complete three complementary tasks designed to capture different aspects of perceptual quality.

**Task A: Overall Preference Ranking.** Participants are shown videos generated by multiple methods and are asked to rank them according to overall visual quality and temporal coherence. Method identities are anonymized and displayed using randomized labels. The presentation order of videos is randomized independently for each participant and each sample. Ranking scores are later converted into numerical scores using a standard rank-based aggregation scheme.

**Task B: Artifact Identification.** Participants are asked to select all videos that exhibit noticeable visual artifacts. Artifact types include ghosting or double exposure, texture flickering, spatial deformation, and inconsistent object boundaries. Multiple selections are allowed for each sample.

**Task C: Motion Plausibility Assessment.** Participants are asked to select videos that contain implausible or inconsistent motion patterns, such as sudden reversals, teleportation-like jumps, or misalignment with the ending keyframe. As in Task B, multiple selections are allowed.

## E. Experimental Setting and Additional Results

### E.1. Experimental Setting

**Evaluation datasets.** We evaluate our method on two widely used benchmarks for generative video inbetweening, covering a diverse range of motion patterns. Specifically, our evaluation set includes 100 video–keyframe pairs from the DAVIS dataset (Pont-Tuset et al., 2017) and 80 pairs selected from the Pexels dataset[1]. These datasets contain videos with complex and large-amplitude motions, such as driving scenarios and human-centric activities.

**Implementation details.** Our method is implemented as an inference-time module and is applied to both parallel and sequential time-reversal samplers, TRF and ViBiD. All experiments are built upon the SVD-XT backbone of Stable Video Diffusion (SVD) (Blattmann et al., 2023) and are executed on a single NVIDIA RTX 4090 GPU. For diffusion sampling, we use the Euler solver with 25 denoising steps, following the default setup of SVD. To enable a controlled and fair comparison with MPD, we adopt the same sampling hyperparameters when integrating our method into TRF and ViBiD, and apply MR-CATR only during the first $0.2T$ sampling steps.

### E.2. Additional Ablation Studies

We conduct a series of ablation studies on the DAVIS dataset to analyze the contribution of each key component.

---

[1] https://www.pexels.com/

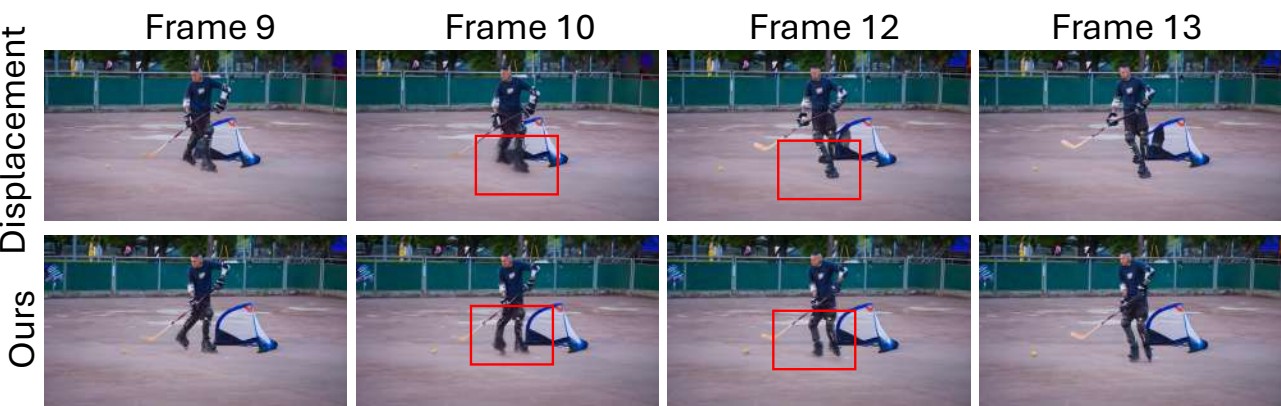

*Figure 8.* Qualitative comparison of alternative motion axis definitions.

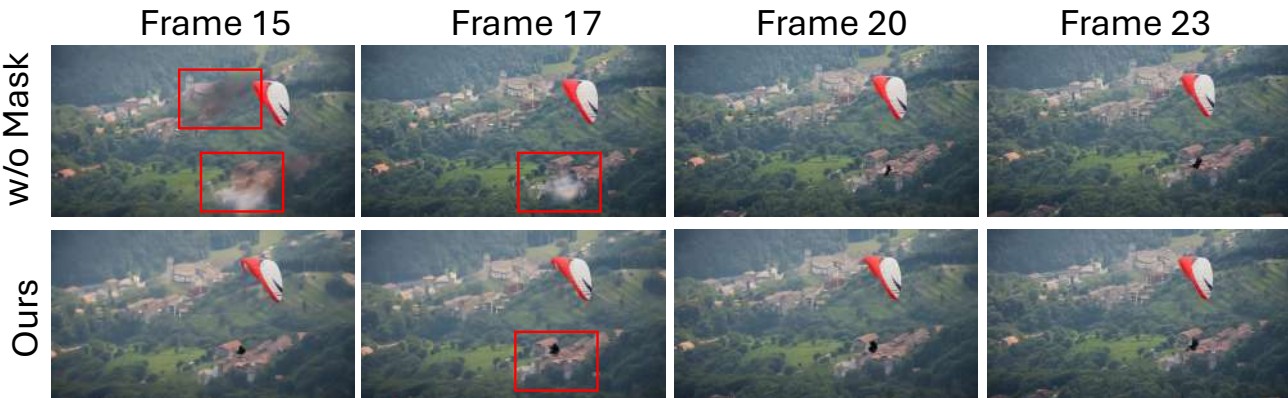

*Figure 9.* Qualitative ablation results of the shared-region masking strategy.

**Alternative motion axis definitions.** We further investigate the effect of different motion axis definitions by replacing the proposed denoised latent residuals with simpler motion cues. Specifically, we consider an alternative displacement-based axis that is constructed solely from the global start–end latent displacement, without leveraging frame-to-frame denoised residual information. Empirically, the displacement-based axis performs reasonably on sequences with near-linear motion but struggles to capture complex and non-linear motion patterns. As shown in Table 7, relying only on a global displacement leads to inaccurate motion directions and degraded temporal consistency. The qualitative results in Figure 8 further show that the former fails to properly capture nonlinear motion (e.g., turning) between Frames 10 and 12. In contrast, the residual-based motion axis consistently yields more stable and temporally coherent results. These observations indicate that denoised latent residuals provide more informative motion cues prior to motion modeling, as they naturally reflect the internal motion representations learned by the I2V diffusion model.

**Effect of shared-region masking.** We evaluate the effect of the shared-region masking strategy, which restricts motion alignment to regions that are visible in both the starting and ending keyframes. As shown in Table 8, such global alignment tends to over-constrain regions without reliable correspondences, leading to degraded temporal consistency. As illustrated in Figure. 9, the former incorrectly blurs the background (e.g., Frames 15 and 17). In contrast, restricting motion alignment to the shared regions yields more stable and clearer results, providing sufficient flexibility when generating scenes with complex backgrounds. These results demonstrate that shared-region masking plays an important role in improving the robustness of motion alignment, particularly in complex real-world scenarios where the keyframes exhibit only partial overlap.

| Method | Motion Axis | LPIPS↓ | FID↓ | FVD↓ |
|---|---|---|---|---|
| Ours+TRF | Displacement-based | 0.314 | 60.2 | 597.1 |
| | Residual-based (Full) | **0.228** | **35.5** | **523.6** |
| Ours+ViBiD | Displacement-based | 0.342 | 58.8 | 602.4 |
| | Residual-based (Full) | **0.210** | **36.9** | **519.5** |

*Table 7.* Ablation study on alternative motion axis definitions.

| Method | Alignment Region | LPIPS↓ | FID↓ | FVD↓ |
|---|---|---|---|---|
| Ours+TRF | Global (w/o Mask) | 0.237 | 37.6 | 527.7 |
| | Shared-region (Full) | **0.228** | **35.5** | **523.6** |
| Ours+ViBiD | Global (w/o Mask) | 0.232 | 37.2 | 522.6 |
| | Shared-region (Full) | **0.210** | **36.9** | **519.5** |

*Table 8.* Ablation study on shared-region masking.

## F. Visualization of Residual-Guided Alignment

We further visualize how the backward update is aligned by MR-CATR in decoded frame space. Given the start-conditioned clean prediction $\hat{x}_{0|t,c_{\text{start}}}$, the current base latent $x_{\text{base}}$, and a representative frame index $k$, we compare three residual maps:

$$R_{\text{start}}^{(k)} = \left| D\left(\hat{x}_{0|t,c_{\text{start}}}^{(k)}\right) - D\left(\hat{x}_{0|t,c_{\text{start}}}^{(k-1)}\right)\right|, \tag{37}$$

$$R_{\text{raw}}^{(k)} = \left| D\left(x_{\text{base}} + d_t^{\text{bwd}}\right)^{(k)} - D(x_{\text{base}})^{(k)}\right|, \tag{38}$$

and

$$R_{\text{align}}^{(k)} = \left| D\left(x_{\text{base}} + d_t^{\text{bwd-aligned}}\right)^{(k)} - D(x_{\text{base}})^{(k)}\right|, \tag{39}$$

where $D(\cdot)$ denotes the latent decoder and

$$d_t^{\text{bwd-aligned}} = \langle d_t^{\text{bwd}}, m_t\rangle m_t. \tag{40}$$

Figure 10 shows the start residual, the raw backward residual, and the aligned backward residual at the same reverse step. The raw backward residual contains scattered off-axis changes, reflecting the conflict between the start- and end-conditioned motion priors. After projection, the aligned backward residual suppresses these noisy components while preserving the dominant residual structure, resulting in a pattern that is more consistent with the start-conditioned motion residual. This confirms that the projection in MR-CATR is not merely a vector-space abstraction, but also produces a more coherent residual pattern after decoding to frame space.

### F.1. Backbone Generalization

Our main experiments use SVD-XT as the backbone because the training-free baselines compared in this paper, including TRF, ViBiD, and MPD, are all commonly instantiated in the same SVD setting. This controlled setup allows us to isolate the effect of the sampling strategy rather than mixing it with differences in backbone capacity. Nevertheless, MR-CATR is not tied to SVD or to a U-Net architecture. Recent large-scale video diffusion models such as Wan (Wan et al., 2025) and CogVideoX (Yang et al., 2025) adopt diffusion-transformer-based designs, while still providing condition-specific denoised predictions or equivalent reverse proposals that can be used by sampler-level methods.

Formally, let $f_\theta$ be a generic conditional video diffusion backbone, which may be implemented by either a U-Net or a DiT. At reverse step $t$, the backbone output under condition $c$ can be converted into a clean latent estimate by a parameterization-specific transformation:

$$y_{t,c} = f_\theta(x_t, t, c), \qquad \hat{x}_{0,c} = \mathcal{T}_t(x_t, y_{t,c}), \tag{41}$$

where $\mathcal{T}_t$ depends on the diffusion parameterization, but not on the internal network architecture. For example, EDM/data-prediction models directly produce $\hat{x}_{0,c}$, while $\epsilon$-prediction models recover it from the predicted noise.

MR-CATR only operates on these output-level quantities: the start- and end-conditioned clean estimates, their corresponding reverse proposals, and the temporal reversal operator. The motion priors are computed from denoised temporal residuals,

$$m_t^{\text{start}} = \mathcal{M}(\hat{x}_{0,c_{\text{start}}}), \qquad m_t^{\text{end}} = \mathcal{M}(R(\hat{x}_{0,c_{\text{end}}})), \tag{42}$$

Table 9. Sanity check of MR-CATR on the DiT-based Wan-I2V backbone. Lower is better.

| Method | DAVIS FVD↓ | DAVIS LPIPS↓ | Pexels FVD↓ | Pexels LPIPS↓ |
|---|---|---|---|---|
| Wan-I2V + MR-CATR on TRF | 512.8 | 0.159 | 461.4 | 0.162 |
| Wan-I2V + MR-CATR on ViBiD | **506.1** | **0.156** | **459.6** | **0.157** |

Table 10. Absolute optical-flow magnitude analysis. $M_g$ denotes the average motion magnitude of generated videos, and $r_{\mathrm{mag}}$ is normalized by the ground-truth magnitude. Values closer to the ground truth indicate more realistic motion strength.

| Method | $M_g$ | $r_{\mathrm{mag}}$ |
|---|---|---|
| GT | 26.04 | 1.00 |
| TRF | 19.72 | 0.76 |
| MR-CATR on TRF | 23.18 | 0.89 |
| ViBiD | 20.53 | 0.79 |
| MR-CATR on ViBiD | **23.96** | **0.92** |

where $\mathcal{M}(\cdot)$ denotes the residual aggregation and normalization in Sec. 3.1, and $R(\cdot)$ denotes temporal reversal. The backward update is then aligned and fused at the sampler level:

$$d_t^{\text{bwd-aligned}} = \langle d_t^{\text{bwd}}, m_t \rangle m_t, \qquad x_{t-1} = x_t + (1 - \beta_t) d_t^{\text{fwd}} + \beta_t d_t^{\text{bwd-aligned}}. \tag{43}$$

Thus, MR-CATR does not rely on U-Net-specific skip connections, convolutional inductive bias, attention-map editing, or SVD-specific hidden features; it only requires denoised predictions and reverse proposals from the backbone-sampler pair.

To verify this beyond SVD, we conduct a sanity check on a DiT-based I2V backbone, Wan-I2V. As shown in Table 9, MR-CATR remains applicable to the stronger DiT backbone and obtains competitive results when plugged into both TRF and ViBiD-style samplers. This supports our view that MR-CATR is a sampler-level and output-level method rather than an architecture-specific modification.

### F.2. Motion Magnitude Analysis

To directly examine whether MR-CATR suppresses motion magnitude, we compute the average inter-frame optical-flow magnitude of generated videos. For a video $V = \{I_1, \ldots, I_N\}$, let

$$F_i(p) = \text{Flow}(I_i, I_{i+1})(p), \tag{44}$$

where $p \in \Omega$ indexes image-grid locations and $\text{Flow}(\cdot, \cdot)$ is a fixed RAFT optical-flow estimator (Teed & Deng, 2020). We define the average motion magnitude as

$$\mathcal{M}(V) = \frac{1}{(N-1)|\Omega|} \sum_{i=1}^{N-1} \sum_{p \in \Omega} \|F_i(p)\|_2. \tag{45}$$

For each method, we report the generated motion magnitude $M_g = \mathcal{M}(V_{\text{gen}})$ and the relative ratio

$$r_{\mathrm{mag}} = \frac{M_g}{M_r}, \qquad M_r = \mathcal{M}(V_{\text{gt}}). \tag{46}$$

Here, $r_{\mathrm{mag}} \approx 1$ indicates motion magnitude close to the ground truth, while $r_{\mathrm{mag}} < 1$ indicates weaker motion.

As shown in Table 10, both TRF and ViBiD produce relatively conservative motion compared with the ground truth, which is expected when forward and backward trajectories conflict. MR-CATR does not further suppress motion magnitude; instead, it increases $r_{\mathrm{mag}}$ from 0.76 to 0.89 on TRF and from 0.79 to 0.92 on ViBiD. This suggests that resolving early bidirectional motion conflict helps the sampler maintain a stronger and more faithful motion layout throughout the remaining denoising process.

### F.3. More Qualitative Results

We present additional qualitative comparisons in Figures 11–14.

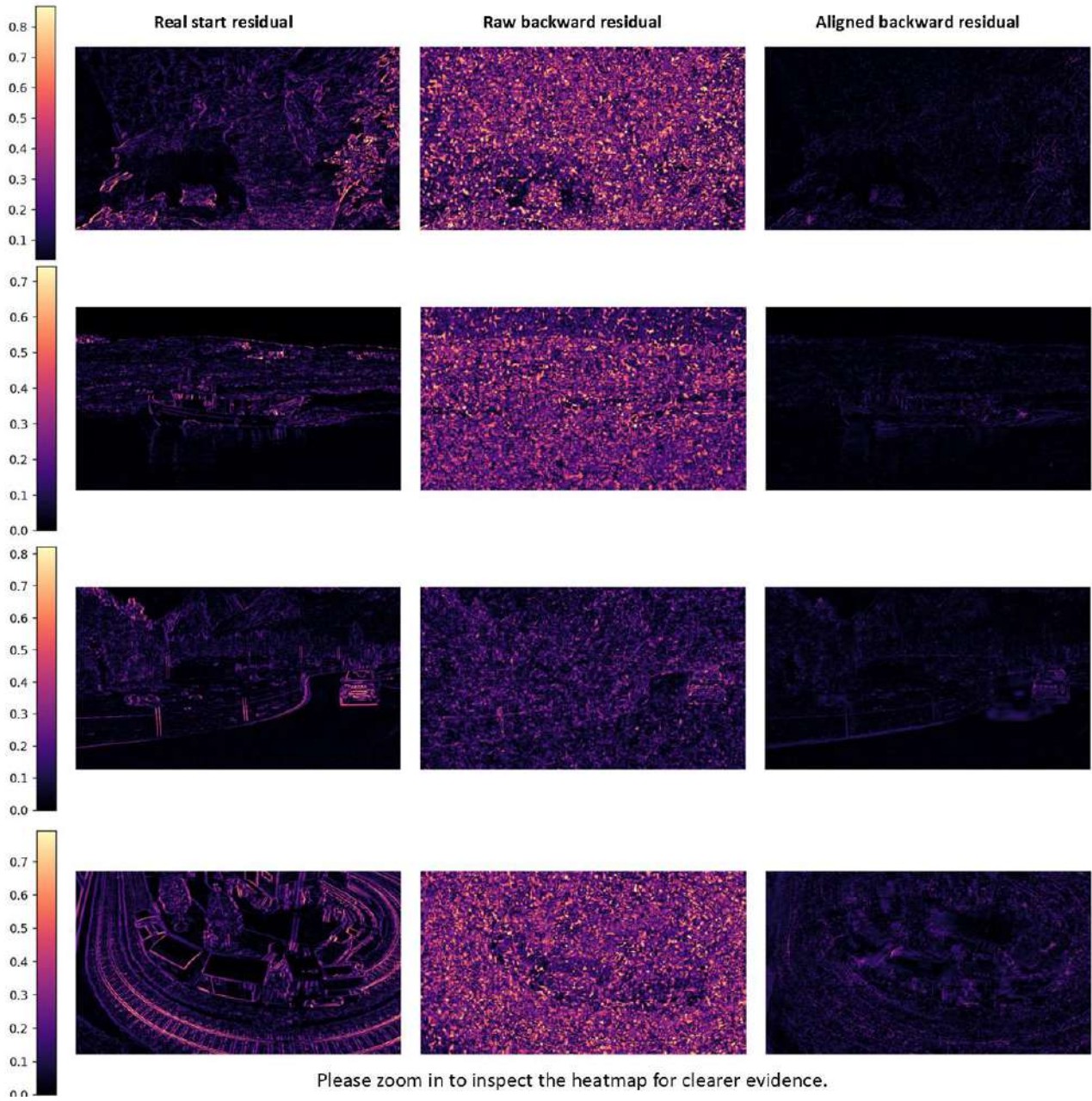

*Figure 10.* Visualization of residual-guided alignment. We compare the start-conditioned residual $R_{\mathrm{start}}$, raw backward residual $R_{\mathrm{raw}}$, and aligned backward residual $R_{\mathrm{align}}$ at the same reverse step. The aligned residual suppresses scattered off-axis changes and better follows the dominant motion structure.

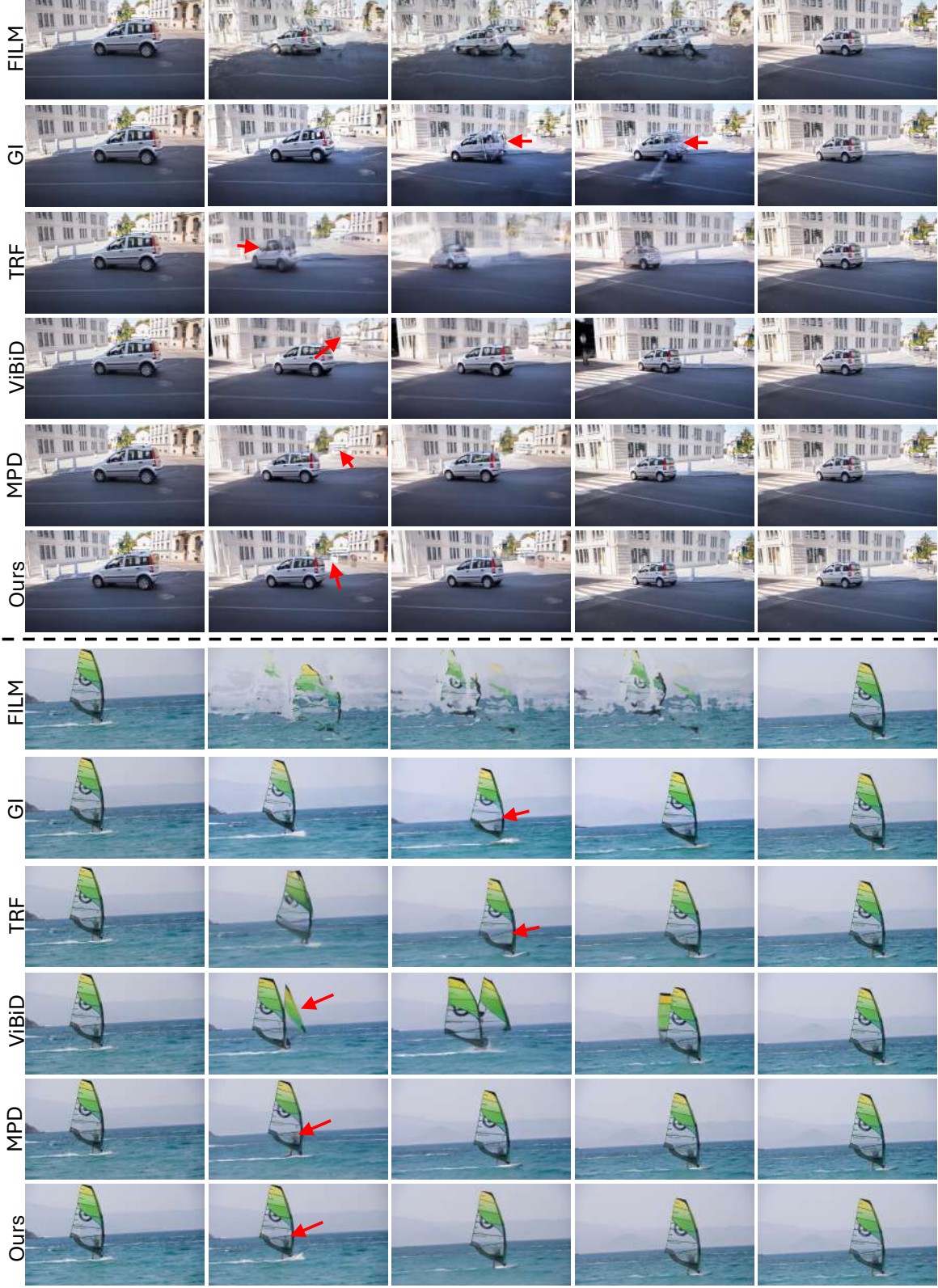

*Figure 11.* More qualitative comparison with existing methods.

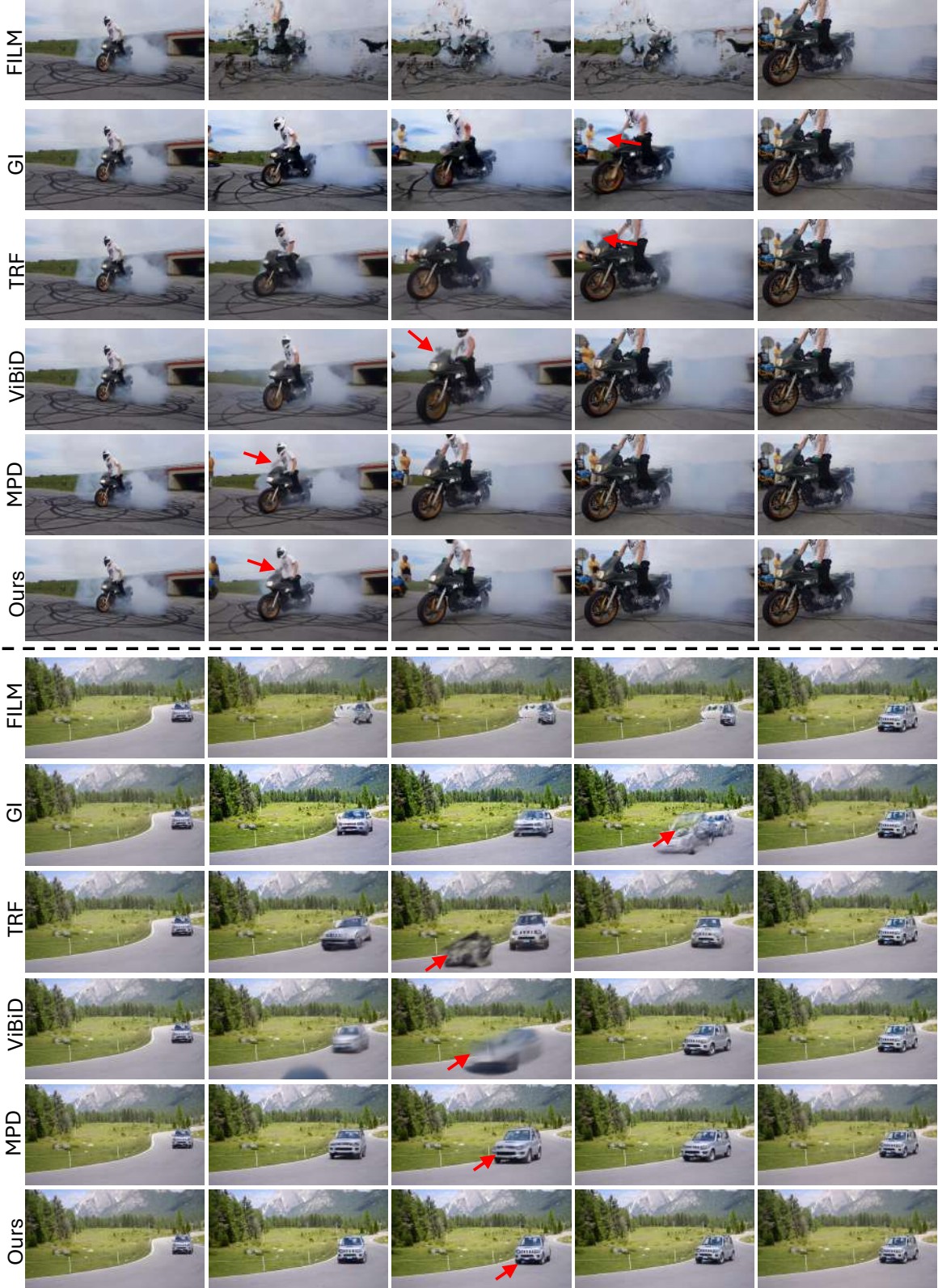

*Figure 12.* More qualitative comparison with existing methods.

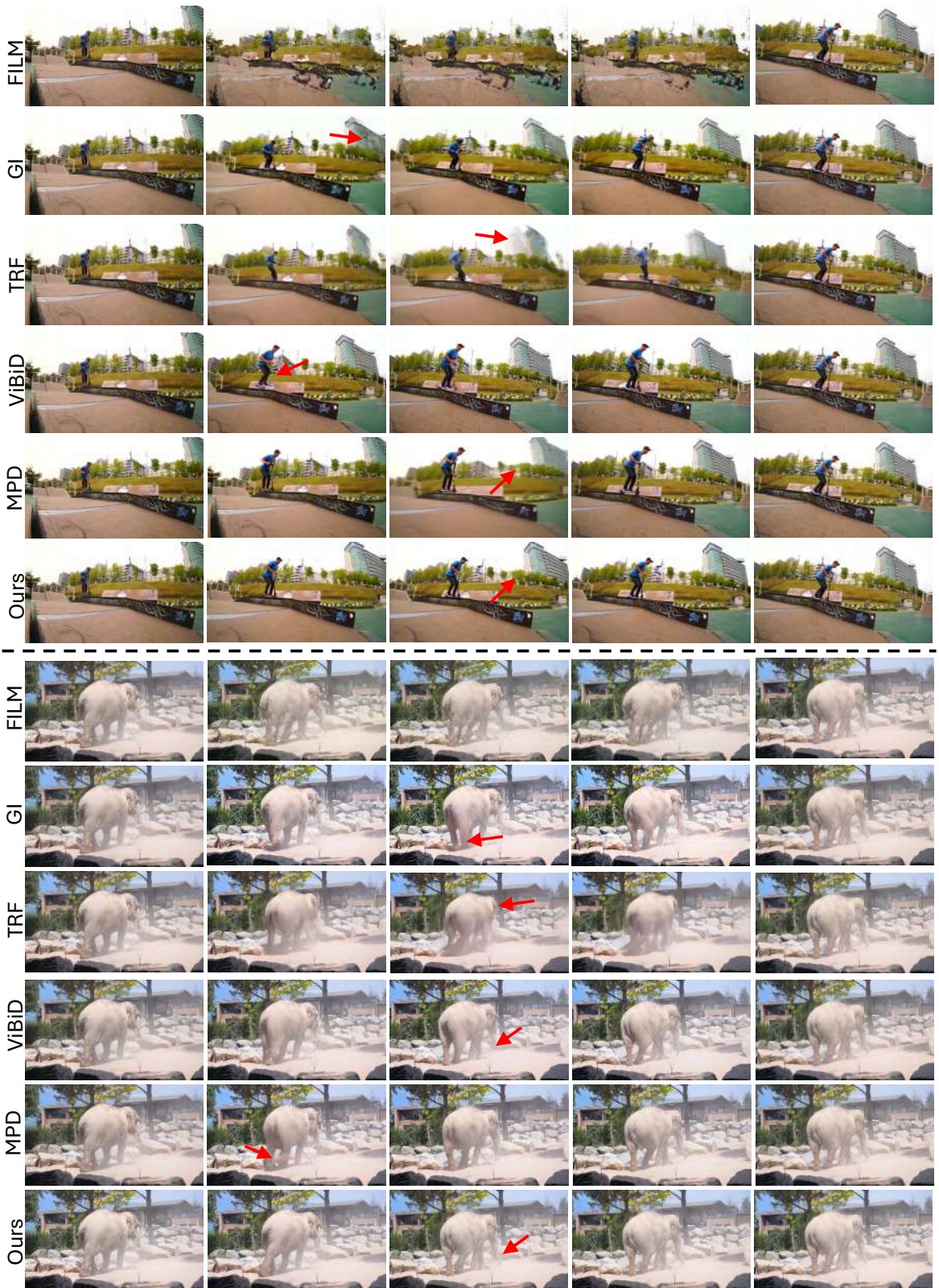

*Figure 13.* More qualitative comparison with existing methods.

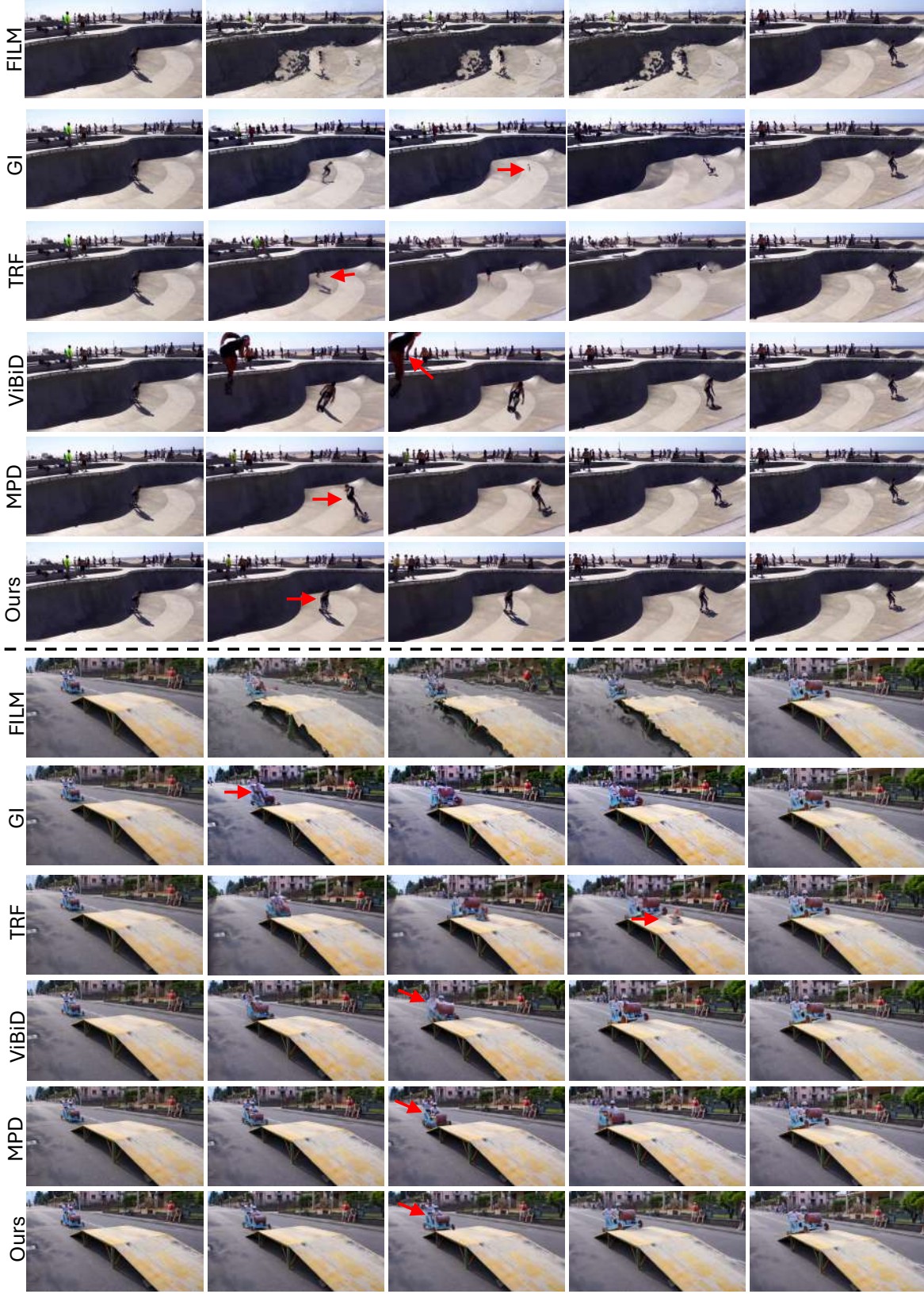

*Figure 14.* More qualitative comparison with existing methods.

