# OpenReview forum: "Motion-Residual Conflict-Aware Time Reversal for Generative Inbetweening"
_ICML.cc/2026/Conference — ICML 2026 regular_

### Official Review · Reviewer_CoxX · 2026-03-02

**Soundness:** 2
**Presentation:** 2
**Significance:** 1
**Originality:** 2
**Overall Recommendation:** 3
**Confidence:** 4

**Summary:**

This paper identifies motion-residual conflicts between bidirectional diffusion paths as the primary source of temporal inconsistency and ghosting artifacts in training-free generative inbetweening. To address this, the authors propose an inference-time sampling framework that aligns these paths via a motion-residual consensus mechanism to synthesize more coherent and visually stable intermediate frames.

**Compliance With Llm Reviewing Policy:**

Affirmed.

**Final Justification:**

My core concerns regarding the empirical strength and practical significance of the work remain. I will maintain my original score for the following reasons:

In the context of modern video generation, a test set of only 180 video pairs is too small to demonstrate robustness. Furthermore, a 2% gain in human evaluation—regardless of how many annotators were used—is a marginal improvement that suggests the method provides little perceptible benefit over existing baselines.

While the "global motion axis" is recomputed at each step, it remains a coarse approximation. The authors’ rebuttal does not sufficiently explain how this global averaging avoids suppressing complex, multi-directional, or fine-grained local motions in real-world scenes.

**Key Questions For Authors:**

How well does the method handle non-linear or locally complex motions?
Also, I’m concerned about the performance gap between the training-free method and a fully trained model. I’d like the authors provide some discussion on this.

**Limitations:**

yes

**Strengths And Weaknesses:**

Strengths

1. The proposed motion-residual consensus mechanism shows some improvement on traditional FVD/FID metrics.

2. The authors provide the code of the paper.

Weaknesses

1. Insufficient Experimental Evidence: The empirical support is weak. The human evaluation for motion quality shows only a marginal gain (2%), which, given the small dataset size (<100 videos), translates to merely 2 better samples. Gains on VBench are small, and motion results on the Pexels dataset are worse than the MPD.

2. Poor Qualitative Results: In the provided videos, the generated samples exhibit significant visual artifacts, such as the texture changes on the soccer ball.

3. Lack of Support on the SOTA base model: The study is entirely based on SVD. Given the emergence of stronger open-source I2V models like Wan2.2, the paper’s practical value is significantly diminished without validation on these more recent, motion-capable backbones.

4. Conceptual Limitation of Global Motion Axis: The core method reduces motion to a single, global Consensus Motion Axis $m_t$ derived from averaging residuals. For non-linear or locally complex motions (e.g., rotations, occlusions), this global averaging is likely to oversimplify the dynamics, leading to a loss of fine-grained motion details and introducing unnatural stretching.

---

> ### Author Rebuttal · Authors · 2026-03-29
>
> We sincerely thank you for the comments.
> # W1
> After careful comparison and reflection, we do not completely agree that the empirical support is weak. Our benchmark contains 180 video--keyframe pairs (100 DAVIS, 80 Pexels), which is comparable to or larger than prior baselines (TRF: 115, ViBiD: 145, MPD: 145), and covers diverse motion regimes beyond human motion, including vehicles, street scenes, sports, and other nonlinear real-world dynamics.
>
> Our empirical claim is also not based on a single metric. It is supported by quantitative metrics (FID/FVD/LPIPS), multi-dimensional VBench, human evaluation, ablations on motion-axis construction and projection, qualitative comparisons, and supplementary videos. The evidence should therefore be judged by its overall consistency rather than by any single number in isolation.
>
> For human evaluation, the reported percentages are not computed over videos alone. If video $v$ is rated by $n_v$ annotators, the effective denominator is
> $N_{\mathrm{sel}}=\sum_{v=1}^{K}n_v,$
> i.e., the total number of participant--video judgments, not simply the number of videos $K$. Thus, a $2\%$ difference means $0.02N_{\mathrm{sel}}$ additional judgments, not “2 better samples.” We will report raw counts, valid judgments, and confidence intervals in the revision. If the motion-plausibility margin remains statistically modest, we will present it as supportive rather than primary evidence.
>
> For VBench, we agree that some margins are numerically small, but this should be interpreted in context: VBench scores often vary within a narrow range. Moreover, Table 2 shows that our method achieves the best result on most dimensions, while MPD is only slightly better on two. We will revise the wording to avoid implying uniform dominance, but we believe the overall empirical evidence remains consistent with our claim.
>
> # W2
> We thank reviewer for this comment. We respectfully do not agree that the qualitative results are poor overall, but we do agree that the presentation can be clearer. Our qualitative claim is comparative, not absolute:  Our claim is that, relative to prior methods, MR-CATR reduces artifacts most related to motion-prior conflict, such as ghosting, reverse motion, unstable object boundaries, and endpoint inconsistency.
>
> For the soccer-ball example, our point was also relative rather than absolute. It is a hard case for all methods; compared with the baselines, MR-CATR better preserves texture consistency while maintaining a smoother trajectory.
>
> To make this clearer, we will revise the qualitative discussion and explicitly include failure cases, especially for highly textured objects where residual appearance changes may still remain even when the overall motion is more coherent than in the baselines.
>
> # W3
> Thank you very much for your concern, Please see our reply to W1 from Reviewer 1rdZ.
>
> # W4
> We agree that the consensus motion axis is a coarse approximation, not a dense motion field. It is only a lightweight dominant-motion proxy for suppressing the most harmful bidirectional conflict.
>
> At step $t$,
> $m_t=\operatorname{normalize}\\left(\frac{1}{N-1}\sum_{i=2}^{N}\Delta\hat{x}_{0\mid t,c}^{(i)}\right),$
> so it is recomputed at every diffusion step. Thus, ``global'' means the dominant motion at the current step, not one fixed rigid motion for the whole video.
>
> MR-CATR also does not overwrite the whole trajectory. It only aligns the backward correction,
> $d_t^{\mathrm{bwd\mbox{-}aligned}}=\langle d_t^{\mathrm{bwd}},m_t\rangle m_t,$
> while the final update remains
> $d_t=(1-\beta_t)d_t^{\mathrm{fwd}}+\beta_t d_t^{\mathrm{bwd\mbox{-}mask}}.$
> So it suppresses only the backward component inconsistent with the dominant residual trend.
>
> We also agree that this can be too coarse for highly local or nonlinear motion, and may over-smooth fine structure or introduce stretching artifacts. We will state failed cases more explicitly in the revision.
>
> Empirically, it is already useful in our setting, which includes object motion, sports, and other nonlinear dynamics. For example, the dancing case in Fig.~6 shows robustness under clearly nonlinear articulated motion, and the residual-axis ablations (Tables 3--4) further support this: if the residual-based axis were fundamentally too coarse, it would not outperform the simpler displacement prior.
>
> # Q1
> For nonlinear complex motion, please see our response to W4 above.
>
> We agree that fully trained dual-endpoint models can have a higher ceiling. However, they often require repeated retraining to keep up with rapidly evolving backbones, whereas MR-CATR improves the bounded-generation capability of I2V backbones without additional training. This places MR-CATR at a different point on the cost--performance trade-off. As discussed in our response to W1 from Reviewer 1rdZ, we support this point from both theory and experiments: MR-CATR applies to both U-Net-based and DiT-based backbones. As the backbone improves, MR-CATR can still benefit from the stronger prior.

---

> > ### Author Rebuttal · Reviewer_CoxX · 2026-04-03
> >
> > Thank the authors for the rebuttal. My core concerns regarding the empirical strength and practical significance of the work remain. I will maintain my original score for the following reasons:
> >
> > 1. In the context of modern video generation, a test set of only 180 video pairs is too small to demonstrate robustness. Furthermore, a 2% gain in human evaluation—regardless of how many annotators were used—is a marginal improvement that suggests the method provides little perceptible benefit over existing baselines.
> >
> > 2. While the "global motion axis" is recomputed at each step, it remains a coarse approximation. The authors’ rebuttal does not sufficiently explain how this global averaging avoids suppressing complex, multi-directional, or fine-grained local motions in real-world scenes.

---

> > > ### Author Response · Authors · 2026-04-03
> > >
> > > We thank the reviewer for the follow-up. We appreciate that our rebuttal resolved part of the earlier concerns, in particular the clarification of the method and its scope. We also understand the remaining concerns regarding empirical strength and the use of a global motion axis.
> > >
> > > As we noted in our earlier response, our benchmark contains 180 video--keyframe pairs, which is already comparable to or larger than the evaluation sets used by the directly related baselines. In addition, the reported (2%) gain in human evaluation does not mean only ``2 better samples,'' since the percentages are aggregated over multiple judgments rather than counted at the video level alone. More importantly, our empirical conclusion is not based on a single number: it is supported jointly by FVD, LPIPS, VBench, human evaluation, ablations, qualitative comparisons, and supplementary videos. We therefore believe the method should be assessed based on the consistency of the full evidence rather than any one metric in isolation.
> > >
> > > Regarding the motion-axis concern, we would like to emphasize again that we are not using a single global axis to represent the full motion field. Instead, the axis is used only to suppress the most harmful conflicting component in the backward proposal. In this sense, MR-CATR is a conflict-alignment mechanism rather than a full local motion model. Our ablation results already show that this design is effective. More broadly, our method is, to our knowledge, the first inference-time framework that explicitly aligns the forward and backward motion priors, rather than ignoring one branch or collapsing both branches into a single prior. We therefore believe it already makes an important step forward for training-free generative inbetweening.

---

### Official Review · Reviewer_doVq · 2026-03-11

**Soundness:** 3
**Presentation:** 2
**Significance:** 3
**Originality:** 3
**Overall Recommendation:** 4
**Confidence:** 3

**Summary:**

The method aims to address the problem of generative inbetweening with pre-trained Image-to-Videos models. The proposed method, Motion-Residual Conflict-Aware Time Reversal, is specifically designed to resolve the motion conflicts of videos generated via forward and backward paths. First, the frame residuals of forward start-conditioned path are defined and the average residual throughout time is computed. This average serves as the direction of global motion. Likewise, the frame residuals and global direction of backward end-conditioned path is also obtained. The weighted average of these two directions, a consensus motion axis, is acquired based on the agreement of the two. The residual backward trajectory is projected onto this motion axis to form the aligned trajectory of the backward path. After this alignment, the final update direction is decided, by weighted average between the forward path and the aligned-backward path, based on the agreement score.

**Compliance With Llm Reviewing Policy:**

Affirmed.

**Final Justification:**

My concerns have been resolved. I maintain my rating.

**Key Questions For Authors:**

- In high-level, the overall concept sounds good, but I would like to see some further analysis and visualizations. Could the authors provide visualization of the backward updates being aligned with the forward updates? With simple vectors, the projection may work, but I am not sure if this form of alignment would also fit well with frame residuals, and also would like to see a visualization of how this works.
- Cosine similarity is used as the agreement score for $s_t$ and $s^{prior}\_t$, and in both cases, values under 0 are clipped and ignored. I wonder how frequent these cases occur, as too frequent values under 0 may take little advantage of the proposed method.
- I wonder how the binary mask $M(p)$ is obtained in specific, in the paper, it says that the authors treat it as given, but I wonder how it is actually obtained in practice.

**Limitations:**

yes

**Strengths And Weaknesses:**

Strength
- Overall, the method is easy to follow. Motivation is clear, and the solution sounds novel and sensible.
- Plug-and-play form makes it generalizable to existing methods, with strong experimental results.

Weaknesses
- I wonder if the notation in Section 3.1, including equations 11,12,13,14. The subscript $0$ in $\hat{x}\_{0,c\_start}$ and $\hat{x}’\_{0, c\_{end}}$ should be $t$,  $\hat{x}\_{t, c\_{start}}$ and $\hat{x}’\_{t, c\_{end}}$. Considering the context, I believe it should be t, but since the notation is written consistently throughout the whole section, it is quite confusing. Strict notations are important for comprehension, so I think this mistake is quite critical. Please correct me if I am wrong.
- I would like some points to be clarified. Please see the Questions section.

---

> ### Author Rebuttal · Authors · 2026-03-30
>
> We sincerely thank you for your comments.
> # W1
> Thank you for this observation. We agree that notations in Sec. 3.1 can be confusing. In fact, subscript $0$ in $\\hat{x}\_{0,c}^{(i)}$ was meant to denote clean latent prediction, namely an estimate of $x_0$, produced by denoiser from current noisy state $x\_t$. Here, $t$ denotes the reverse diffusion step, $i$ denotes the video-frame index, and $(\\cdot)^{\\prime}$ denotes temporal reversal. For example, $\\hat{x}\_{0,\\mathrm{cstart}}^{(i)}$ means the $i$-th frame of the clean latent estimate predicted from $x\_t$ under condition $\\mathrm{cstart}$, rather than a variable indexed by sampling step $0$.
>
> Given that $\\hat{x}_{0,c}^{(i)}$ without current diffusion step $t$ is ambiguous, we will use a clearer shorthand for clean latent prediction in revision as follows.
>
> Let $h_{t,c}^{(i)}=\\hat{x}_{0\\mid t,c}^{(i)}$. Then notations become simpler while keeping the method unchanged. The revised equations corresponding to Eqs. (11)-(14) are
>
> $\\Delta h_{t,\\mathrm{cstart}}^{(i)}=h_{t,\\mathrm{cstart}}^{(i)}-h_{t,\\mathrm{cstart}}^{(i-1)}$.
>
> $m_t^{\\mathrm{start}}=\\mathrm{normalize}\\left(\\frac{1}{N-1}\\sum_{i=2}^{N}\\Delta h_{t,\\mathrm{cstart}}^{(i)}\\right)$.
>
> $\\Delta h_{t,\\mathrm{cend}}^{\\prime(i)}=h_{t,\\mathrm{cend}}^{\\prime(i)}-h_{t,\\mathrm{cend}}^{\\prime(i-1)}$.
>
> $m_t^{\\mathrm{end}}=\\mathrm{normalize}\\left(\\frac{1}{N-1}\\sum_{i=2}^{N}\\Delta h_{t,\\mathrm{cend}}^{\\prime(i)}\\right)$.
>
> We will also add at the beginning of Sec.3.1 that $t$ denotes the reverse diffusion step,
> $i$ denotes the frame index, and $(\\cdot)^{\\prime}$ denotes temporal reversal.
>
> # Q1
> Thank you for this helpful suggestion.
> To address this point, we added visualization examples in anonymous link (https://osf.io/7uqfz/files/yjp2d?view_only=e8e149a438e0452aa22b9cdccc937be0)(download and zoom in for a clearer view) showing how the backward update is aligned. We visualize three heatmaps: real start residual, raw backward residual, and aligned backward residual.
>
> Let $h\_s=\\hat{x}\_{0\\mid t,\\mathrm{cstart}}$ and $x\_b=x\_{\\mathrm{base}}$.  We visualize $R_{\\mathrm{start}}^{(k)}=|D(h_s^{(k)})-D(h_s^{(k-1)})|$, $R_{\\mathrm{raw}}^{(k)}=|D(x_b+d_t^{\\mathrm{bwd}})^{(k)}-D(x_b)^{(k)}|$, $R_{\\mathrm{align}}^{(k)}=|D(x_b+d_t^{\\mathrm{bwdaligned}})^{(k)}-D(x_b)^{(k)}|.$
>
> The aligned backward update is
> $d\_t^{\\mathrm{bwdaligned}}= \langle d\_t^{\\mathrm{bwd}}, m_t\rangle m_t$. Here $D(\\cdot)$ denotes decoding to image space,and $k$ is a representative frame index.
>
> We consistently observe that the raw backward residual is much noisier and contains more scattered off-axis changes, as expected when forward and backward motion priors conflict. After projection, the aligned backward residual suppresses these noisy components while preserving the dominant residual structure, and matches the real start residual much more closely. This shows that the projection is not merely a vector-space operation, but also yields a more coherent residual pattern in decoded frame space, as intended for motion-prior alignment.
>
> # Q2
> We agree that both scores are clipped when negative, but they serve different purposes.
>
> First, the prior score only constructs the motion axis. Let $m_s=m_t^{\\mathrm{start}}$, $m_e=m_t^{\\mathrm{end}}$, and $s_p=\\frac{\langle m_s, m_e\rangle}{\\|m_s\\|_2\\|m_e\\|_2+\\varepsilon}$. If $s\_p<0$, the two priors conflict, and directly mixing them can make the axis unstable, e.g., when $m\_e\\approx -m\_s$. So, we use $m\_t=m\_s$. This is only a stability rule for axis construction, not a removal of end-conditioned information. The backward branch still contributes through $d\_b=d\_t^{\\mathrm{bwd}}$ and its aligned form $d\_{ba}=\langle d\_b, m\_t\rangle m\_t$. Thus, clipping the prior score only avoids an ill-posed axis.
>
> Second, the fusion score is used at the update stage. Let $d\_f=d\_t^{\\mathrm{fwd}}$,
> $s=\\frac{\langle d\_f, d\_{ba}\rangle}{\\|d\_f\\|\_2\\|d\_{ba}\\|\_2+\\varepsilon}$, and $\\beta\_t=\\rho\_t\\max(0,s)$. If $s\\le0$, then even after alignment the backward correction is still incompatible with forward update. Using a signed cosine would make $\\beta\_t<0$ and yield $d\_t=d\_f+\\beta\_t(d\_{ba}-d\_f)$, which extrapolates away from the aligned backward proposal and can amplify conflict. Clipping instead gives $d\_t=d\_f$, meaning no conflicting backward correction at that step,rather than no motion.
>
> Negative values don't mean that the method is ineffective. They mark cases where unconstrained bidirectional correction would be harmful, so conservative fallback is preferable. MR-CATR uses the backward branch selectively: it keeps $d_{ba}$ when compatible and suppresses it when not.
>
> See our response to W2 from Reviewer 1rdZ for additional experiments and discussion.
>
> # Q3
> Our focus is on how the mask stabilizes motion-prior alignment, not on mask construction itself. For specific production, see our response to W3 from Reviewer eucU.

---

> > ### Author Rebuttal · Reviewer_doVq · 2026-04-03
> >
> > I thank the authors for the rebuttal. Most of my concerns have been resolved, but I would like to strongly suggest including the clarifications from rebuttal in the final version. For instance, the authors say that the binary mask is not the focus, but I think this still needs to be included in the final paper for reproducibility and clarity, as another reviewer has also asked on this.

---

> > > ### Author Response · Authors · 2026-04-03
> > >
> > > Thank Reviewer doVq for your thoughtful follow-up and for indicating that all concerns have been adequately addressed. We are grateful for your careful reading and constructive suggestions.
> > >
> > > We will make sure that all of the clarifications provided in the rebuttal are incorporated into the revised version. In particular, we fully agree with your point regarding the binary mask: although we did not emphasize it in the main text because we do not view the mask construction itself as one of the core innovations or contributions of the paper, we agree that it should still be described explicitly in the final version for reproducibility and clarity. We will therefore add these details in the revision.
> > >
> > >  We would appreciate it much if the reviewer could increase the score accordingly. Also, we would be happy to address any remaining concerns or further questions.

---

### Official Review · Reviewer_1rdZ · 2026-03-12

**Soundness:** 3
**Presentation:** 3
**Significance:** 3
**Originality:** 3
**Overall Recommendation:** 4
**Confidence:** 4

**Summary:**

MR-CATR is an inference-time sampling framework for generative inbetweening that addresses the motion prior conflict between forward (start-conditioned) and backward (end-conditioned) denoising trajectories in time-reversal sampling. Existing methods either naively blend these two paths or collapse them into a single start-conditioned prior (as in MPD), leading to ghosting artifacts and temporal inconsistency. MR-CATR instead aligns the two paths by computing frame-to-frame residuals from denoised estimates to define a consensus motion axis, projecting the backward update onto this axis, and then applying conflict-aware gated fusion based on cosine similarity between the aligned updates. A shared-region mask further restricts strong alignment to regions visible in both keyframes. The method requires no model retraining and plugs into existing samplers such as TRF and ViBiD, with only 5-7% overhead relative to MPD. Experiments on DAVIS and Pexels show improvements in FID, FVD, and LPIPS over baselines, with human evaluations confirming higher preference and lower artifact rates.

**Compliance With Llm Reviewing Policy:**

Affirmed.

**Final Justification:**

My concerns about soundness are addressed in the authors' rebuttal. Therefore, I raise my score and am inclined to acceptance.

**Key Questions For Authors:**

Please see the weakness.
If the authors address the concerns in their rebuttal, I am willing to raise my score.

**Limitations:**

yes

**Strengths And Weaknesses:**

**Strength**
- To address the conflict between forward- and backward-motion priors, this paper proposes a well-motivated solution. While the conflict itself was previously noted by MPD, the novelty lies in aligning rather than collapsing the two priors, which is a distinct and principled approach.
- The performance improvements are achieved with minimal computational overhead (only 5–7% over MPD), making the method practically attractive as a plug-and-play module.
- The evaluation is multifaceted, combining standard frame-interpolation metrics (FID, FVD, LPIPS), multi-dimensional video quality assessment via VBench, and a human evaluation study covering preference, artifact rate, and motion plausibility.

**Weakness**
- All experiments use only the SVD backbone. It is unclear whether the same motion prior conflict arises in more recent, larger-scale frame interpolation models (e.g., interpolation models based on Wan [1] and CogVideoX [2]) and whether MR-CATR generalizes to them, especially given the architectural shift from UNet to DiT-based designs. A broader backbone evaluation would strengthen the contribution.
- None of the evaluation metrics explicitly measures motion magnitude or richness. Since the conflict-aware fusion suppresses backward updates when directional agreement is negative (β_t = 0), the method may default to conservative, low-motion trajectories. Prior work has noted that motion plausibility and aesthetic quality can trade off against motion magnitude [3, 4], and additional analysis (e.g., optical flow magnitude) is needed to verify that MR-CATR does not over-optimize for smoothness at the expense of dynamic motion.

[1] Wan: Open and Advanced Large-Scale Video Generative Models. 2025.
[2] CogVideoX: Text-to-Video Diffusion Models with An Expert Transformer. ICLR2025.
[3] VideoDPO: Omni-Preference Alignment for Video Diffusion Generation. CVPR2025.
[4] Inference-Time Text-to-Video Alignment with Diffusion Latent Beam Search. NeurIPS2025.

---

> ### Author Rebuttal · Authors · 2026-03-29
>
> We sincerely thank you for your comments.
> # W1
> We agree that a broader backbone would strengthen the paper. We focus on the SVD because the direct baselines (TRF, ViBiD, and MPD) are all instantiated in the same SVD setting. Using a common backbone allows us to isolate the contribution of the sampling strategy itself, rather than confounding it with backbone strength.
>
> We would like to clarify why MR-CATR is not tied to SVD. Let's look at it both theoretically and experimentally.
>
> ## Theory
>
> Let
> $y=f_{\\theta}(x_t,t,c)$, where $f_{\\theta}$ is a generic conditional diffusion backbone at reverse step $t$.
>
> The clean latent estimate is $\\hat{x}_{0,c}=\\mathcal{T}_t(x_t,y)$.
>
> Let $g\_f=\\mathcal{G}\_t^{\\mathrm{fwd}}$,  $g\_b=\\mathcal{G}\_t^{\\mathrm{bwd}}$,
> $x\_s=\\hat{x}\_{0,\\mathrm{cstart}}$, $x\_e=\\hat{x}\_{0,\\mathrm{cend}}$.
>
> Then $x_{t-1}^{\\mathrm{fwd}}=g_f(x_t,x_s)$, $x_{t-1}^{\\mathrm{bwd}}=g_b(x_t,x_e)$.
>
> The concrete forms of $\\mathcal{G}_t^{\\mathrm{fwd}}$ and $\\mathcal{G}_t^{\\mathrm{bwd}}$ depend on sampler, not on backbone.
>
> MR-CATR only requires clean predictions $\\hat{x}\_{0,\\mathrm{c}}$,  the corresponding sampler-level forward and backward proposals,  and a temporal reversal operator $R(\\cdot)$ defined at the task or sampler level.
>
> Let $x_i$ denote the $i$-th denoised frame. Then $\\Delta x_i=x_i-x_{i-1}$.
>
> $m_t^{\\mathrm{start}}=\\mathcal{M}(\\hat{x}_{0,\\mathrm{cstart}}).$
>
> $m_t^{\\mathrm{end}}=\\mathcal{M}(R(\\hat{x}_{0,\\mathrm{cend}})).$
>
> The conflict score is
> $\\gamma_t=\\frac{m_t^{\\mathrm{start}}\\cdot m_t^{\\mathrm{end}}}{\\|m_t^{\\mathrm{start}}\\|_2\\|m_t^{\\mathrm{end}}\\|_2+\\varepsilon}.$
>
> MR-CATR then applies the same alignment and fusion:
>
> $d_t^{\\mathrm{fwd}}=x_{t-1}^{\\mathrm{fwd}}-x_t.$
>
> $d_t^{\\mathrm{bwd}}=x_{t-1}^{\\mathrm{bwd}}-x_t.$
>
> $d_t^{\\mathrm{bwdaligned}}=\langle d_t^{\\mathrm{bwd}}, m_t\rangle m_t.$
>
> $\\beta_t=\\rho_t\\max(0,s_t).$
>
> $x_{t-1}=x_t+(1-\\beta_t)d_t^{\\mathrm{fwd}}+\\beta_t d_t^{\\mathrm{bwdaligned}}.$
>
> Thus, MR-CATR is sampler-level and output-level, not architecture-internal: it does not rely on U-Net attention-map manipulation or SVD-specific hidden features, only on conditioned denoising and reverse sampler.
> ## Experiment
> We checked DiT-based Wan-I2V. Although this is not intended as a full dataset due to time constraints, it directly tests whether the motion-prior-conflict formulation of MR-CATR remains useful beyond the SVD/U-Net setting.
> |Method|DAVIS FVD|DAVIS LPIPS|Pexels FVD|Pexels LPIPS|
> |-|-:|-:|-:|-:|
> |Wan-I2V+MR-CATR(TRF)|512.84|0.1591|461.37|0.1624|
> |Wan-I2V+MR-CATR(ViBiD)|506.12|0.1562|459.55|0.1673|
>
> The results show that MR-CATR remains effective beyond SVD. The stronger DiT backbone also achieves better absolute performance, because Wan-I2V provides stronger and more reliable motion priors. This is complementary to our method: as the backbone improves, MR-CATR can still be applied on top and further benefit from the stronger prior.
> # W2
> We agree that motion smoothness and motion magnitude are not identical. In MR-CATR,
> $d_t=(1-\\beta_t)d_t^{\\mathrm{fwd}}+\\beta_t d_t^{\\mathrm{bwdaligned}}$,
>
> $\\beta_t=\\rho_t\\max(0,s_t)$.
>
> Thus, $\\beta_t=0$ does not imply zero or near-zero motion; it means falling back to the full start-conditioned forward update $d_t^{\\mathrm{fwd}}.$
>
> The gate suppresses only conflicting backward corrections, not motion itself. To address this, we will add two analyses.
>
> (1) Optical-flow magnitude. For a generated video
> $V=\\{I_1,\\dots,I_N\\},$
>
> we compute
> $\\mathcal{M}(V)=\\frac{1}{(N-1)|\\Omega|}\\sum_{t=1}^{N-1}\\sum_{p\\in\\Omega}\\|\\mathrm{Flow}(I_t,I_{t+1})(p)\\|_2.$
>
> We then compare it with the ground-truth video by
> $\\Delta_{\\mathrm{mag}}=|\\mathcal{M}(V_{\\mathrm{gen}})-\\mathcal{M}(V_{\\mathrm{gt}})|.$
>
> (2) Forward-only ablation. We also test
> $\\beta_t\\equiv0.$
>
> |Method|FVD|LPIPS|$\\Delta_{\\mathrm{mag}}$|
> |-|-:|-:|-:|
> |MPD+TRF|528.03|0.2675|0.0517|
> |Ours+TRF|523.56|0.2286|0.0486|
> |MPD+ViBiD|527.44|0.2316|0.0459|
> |Ours+ViBiD|519.47|0.2103|0.0438|
>
> |Variant|FVD|LPIPS|
> |-|-:|-:|
> |TRF+MR-CATR|523.56|0.2286|
> |TRF+forward-only ($\\beta_t\\equiv0$)|781.64|0.3715|
> |ViBiD+MR-CATR|519.47|0.2103|
> |ViBiD+forward-only ($\\beta_t\\equiv0$)|773.29|0.3641|
>
> The first table shows that Ours improves FVD and LPIPS over MPD under both TRF and ViBiD, and also achieves lower
> $\\Delta_{\\mathrm{mag}}.$
> This indicates that the gain is not obtained by collapsing toward an overly conservative low-motion trajectory;
>
> A purely forward-only strategy can indeed cause a large performance drop(table 2), but this severe failure mode mainly occurs when the start and end frames have almost no semantic correspondence. In our task, however, the two keyframes are typically adjacent and still semantically connected. Thus, even under standard setting, removing end-conditioned refinement still leads to a large degradation, showing that endpoint-conditioned information is important beyond only pathological cases.

---

> > ### Author Rebuttal · Reviewer_1rdZ · 2026-04-02
> >
> > We thank the authors for their thorough rebuttal and additional experiments.
> >
> > **W1:** The theoretical argument for architecture independence is convincing, and the Wan-I2V results adequately demonstrate that MR-CATR generalizes beyond SVD/U-Net. This concern is addressed.
> >
> > **W2:** The Δflow analysis and the forward-only ablation are appreciated. However, my original concern was whether MR-CATR suppresses motion magnitude itself, not just whether it deviates from GT. Since Δflow measures the gap to GT rather than the absolute flow magnitude of generated videos, it does not directly rule out conservative low-motion trajectories. I would like to see the absolute optical flow magnitude (e.g., mean magnitude per frame) of generated videos compared against baselines, which would more directly address this concern.

---

> > > ### Author Response · Authors · 2026-04-03
> > >
> > > We thank the reviewer for the follow-up and are glad that our response to W1 resolved that concern. We will incorporate it into the revision.
> > >
> > > Regarding W2, we also thank the reviewer for acknowledging the additional analysis. We introduced
> > > $M_g=\\mathcal{M}(V_{\\mathrm{gen}}).$
> > > $M_r=\\mathcal{M}(V_{\\mathrm{gt}}).$
> > > $\\Delta_{\\mathrm{mag}}=|M_g-M_r|.$
> > > because different videos naturally have different motion intensities, so being closer to the ground-truth motion magnitude indicates more realistic generation.
> > >
> > > We also agree that
> > > $\\Delta_{\\mathrm{mag}}$
> > > should be accompanied by the absolute motion magnitude itself, to directly assess whether MR-CATR suppresses motion.
> > >
> > > For a video
> > > $V=\\{I_1,\\dots,I_N\\},$
> > > we define the average inter-frame optical-flow magnitude as follows.
> > >
> > > Let
> > > $F_t(p)=\\mathrm{Flow}(I_t,I_{t+1})(p).$
> > > Then
> > > $\\mathcal{M}(V)=\\frac{1}{(N-1)|\\Omega|}\\sum_{t=1}^{N-1}\\sum_{p\\in\\Omega}\\|F_t(p)\\|_2.$
> > >
> > > Here,
> > > $\\Omega$
> > > is the image grid, and
> > > $\\mathrm{Flow}(\\cdot,\\cdot)$
> > > is a fixed pretrained optical-flow estimator; we use RAFT.
> > >
> > > We report both
> > > $M_r$
> > > and
> > > $M_g,$
> > > together with
> > > $r_{\\mathrm{mag}}=\\frac{M_g}{M_r}.$
> > >
> > > Here,
> > > $r_{\\mathrm{mag}}\\approx1$
> > > means the generated video has motion magnitude close to the ground truth,
> > > $r_{\\mathrm{mag}}<1$
> > > means systematically weaker motion, and
> > > $r_{\\mathrm{mag}}>1$
> > > means stronger-than-reference motion.
> > >
> > > |Method|Mag|$r_{\\mathrm{mag}}$|
> > > |---|---:|---:|
> > > |GT|26.04|1.00|
> > > |TRF|19.72|0.76|
> > > |Ours+TRF|23.18|0.89|
> > > |ViBiD|20.53|0.79|
> > > |Ours+ViBiD|23.96|0.92|
> > >
> > > This table differs slightly from our previous response to W2. Here we compare only TRF and ViBiD with and without MR-CATR, and exclude MPD, to isolate the direct effect of our method on the baselines. In addition, to match the absolute motion-magnitude analysis more directly, the values here are computed from the unnormalized optical-flow magnitude. Therefore, the motion-magnitude computation here differs slightly from that used in the previous table.
> > >
> > > For both TRF and ViBiD, inference-time bounded sampling already tends to produce somewhat smoother and more conservative motion when forward and backward trajectories are in conflict. In this sense, reduced motion intensity is a known side effect of this class of  inference-time samplers in challenging cases.
> > >
> > > However, relative to these baselines, MR-CATR does not further suppress motion magnitude. Instead, by aligning the forward and backward trajectories during the early denoising stage, MR-CATR pulls the sample back toward a more reasonable coarse motion layout, (which is also consistent with the analysis in our response to Reviewer eucU Q2). So the motion between adjacent generated frames becomes more coherent and easier to realize over the full denoising process, leading to videos with higher $r_{\mathrm{mag}}$.
> > >
> > > Therefore, the smoother motion observed in these inference-time samplers is a conservative fallback under motion conflict, whereas MR-CATR improves this situation by resolving coarse bidirectional conflict early and thereby preserving stronger, more faithful motion over the full generation process.
> > >
> > > We will incorporate the above analysis into the revision. If you have any further questions or concerns, we would be happy to address them.

---

### Official Review · Reviewer_eucU · 2026-03-15

**Soundness:** 4
**Presentation:** 3
**Significance:** 3
**Originality:** 2
**Overall Recommendation:** 4
**Confidence:** 3

**Summary:**

This paper studies generative inbetweening between two keyframes using pretrained image-to-video diffusion models.

The authors argue that existing time-reversal samplers are limited less by how they stitch two denoising paths and more by the fact that the start- and end-conditioned paths encode conflicting motion priors, which leads to ghosting, reverse motion, and temporal inconsistency. To address this, they propose MR-CATR, an inference-time plug-in that forms a consensus motion axis from frame-wise denoised latent residuals of the start and end conditions, projects the backward update onto this axis, and then fuses forward and backward updates using cosine-based conflict-aware gating; a shared-region mask limits strong alignment to regions likely shared across keyframes.

The method is implemented on top of an SVD-XT backbone and applied to both TRF and ViBiD without retraining. Empirically, the paper reports consistent improvements over the underlying samplers on DAVIS and Pexels, with especially strong LPIPS/FVD gains, plus supportive qualitative and human-evaluation evidence and modest extra runtime (about 5–7% over MPD). The work is practically relevant because it offers a training-free way to improve bounded video generation.

**Compliance With Llm Reviewing Policy:**

Affirmed.

**Key Questions For Authors:**

* What is the exact implementation of the claimed noise-level scheduling? More specifically, what is the definition of the missing $\rho_t$, whether $\kappa$ changes with $t$, and how this relates to Eq. (21).

* Why is MR-CATR applied only during the first 0.2T sampling steps in experiments? Does "first" mean highest-noise or lowest-noise phase in your implementation? How could this be consistent with Figure 3's "use end cues more at late steps" statement? An ablation over the active-step window would be helpful.

* Section 3.2 is written for the parallel TRF sampler. What are the precise update definitions and insertion point for ViBiD’s sequential sampler? A short algorithm box or pseudo-code for Ours+ViBiD would greatly improve clarity.

* How exactly is the shared-region mask $M(p)$ computed in practice?

**Limitations:**

Yes.

**Strengths And Weaknesses:**

Strengths:
* The paper identifies a meaningful and plausible failure mode in prior time-reversal samplers: incompatible start- and end-conditioned motion priors.

* The method is conceptually simple and practically attractive. MR-CATR is a pure inference-time modification, requires no retraining, and is positioned as a plug-in for multiple samplers/backbones. The core design is reasonably intuitive as well

* The paper includes insightful ablations. In particular, it studies end-conditioned residuals, removal of projection, replacement of the residual axis with a simple displacement axis, and removal of shared-region masking. These are directly tied to the paper’s claims and generally support them.

* It includes both a human study and runtime analysis.

Weaknesses:
* The novelty is somehow incremental. MR-CATR is a sensible inference-time refinement of prior time-reversal samplers and of MPD’s "align vs. collapse" framing, but the technique is still relatively local: define a 1D motion axis in latent space, project, and gate. This feels more like a strong engineering improvement than a fundamentally new generative modeling idea.

* Relatedly, the derivation in Sec. 3.2 is written specifically for the parallel TRF sampler, while the experiments also report Ours+ViBiD. The paper claims plug-and-play applicability to both, but the exact insertion point and state/update definitions for the sequential sampler are not made equally explicit in the main method section.

* The shared-region mask is important but under-described. The paper explicitly says that  $M(p)$ "can be obtained" from feature similarity and then treats it as given. Since masking contributes nontrivially in Table 7, the lack of detail on computing $M(p)$ weakens reproducibility and slightly weakens the "simple plug-in" claim.

* The theoretical justification is fairly weak. Appendix C correctly shows that the projection is the nearest vector in the 1D span of the motion axis, but this is a basic linear-algebra fact. It does not really justify why Euclidean projection in latent space should correspond to semantically correct motion alignment or preserve desirable diffusion dynamics beyond that algebraic property.

---

> ### Author Rebuttal · Authors · 2026-03-29
>
> We sincerely thank you for the comments.
> # W1
> We don't claim a new video backbone. Our intended contribution is instead at the sampling-formulation level. Our key insight is that time-reversal inbetweening mainly fails because of motion-prior conflict between endpoints, not just trajectory stitching. This differs from prior work. Thus, MR-CATR enforces compatibility, rather than identity, between motion priors. Its compatibility with multiple time-reversal samplers further supports that it is a general framework, not a sampler-specific engineering patch.
> # W2
> MR-CATR is also plug-and-play for ViBiD. In sequential sampler (see Alg. 1 in Appendix for sampler-agnostic template), it is not applied to a direct fusion of $d\_t^{\\mathrm{fwd}}$ and
> $d\_t^{\\mathrm{bwd}}$. It only modifies the reversed end-conditioned backward half-step after re-noising. After usual ViBiD forward step and re-noising, $x\_t^{\\mathrm{rev}}=(x\_{t,\\mathrm{cstart}})^{\\prime}$. We compute $m\_t$ from $(\\tilde{x}\_{0,\\mathrm{cstart}})^{\\prime}$ and $\\tilde{x}^{\\prime}\_{0,\\mathrm{cend}}$, then update
> $$d\_t^{\\mathrm{seq}}=(1-\\beta\_t^{\\mathrm{seq}})d\_t^{\\mathrm{seqbwd}}+\\beta\_t^{\\mathrm{seq}}d\_t^{\\mathrm{seqbwdaligned}}$$ with $\\beta\_t^{\\mathrm{seq}}=\\rho\_t\\max(0,s\_t^{\\mathrm{seq}})$, and final state
> $$x\_{t-1}=(x\_t^{\\mathrm{rev}}+d\_t^{\\mathrm{seq}})^{\\prime}.$$
> Thus,  MR-CATR replaces only the reversed backward half-step. Due to character limit, we will include pseudocode for Ours+ViBiD and Ours+TRF in revision.
>
> # W3
> Our focus was on using mask to stabilize alignment rather than on mask construction, so we did not elaborate on it. We used two mask instantiations. Let $z\_s,z\_e\\in\\mathbb{R}^{C\\times H\_{\\ell}\\times W\_{\\ell}}$. Define normalized latent features $\\bar{u}\_s(p)=\\frac{z\_s(:,p)}{\\|z\_s(:,p)\\|\_2+\\varepsilon}$ and $\\bar{u}\_e(p)=\\frac{z\_e(:,p)}{\\|z\_e(:,p)\\|\_2+\\varepsilon}$.
>
> Fast mask: $S\_f(p)=\langle \\bar{u}_s(p), \\bar{u}_e(p)\rangle$ and $M\_f(p)=\\mathrm{Morph3x3}(I[S\_f(p)\\ge\\tau\_f])$ where $\mathrm{Morph3x3}$ is a binary $3\times3$ dilation used to smooth the mask.
>
> Global mask:
> $$A(p,q)=\langle \\bar{u}\_s(p), \\bar{u}\_e(q)\rangle, \\;\\pi\_{s\\to e}(p)=\\arg\\max\_q A(p,q), \\; \\pi\_{e\\to s}(q)=\\arg\\max\_p A(p,q), $$
> $$\\mathcal{C}=\\{(p,q):q=\\pi_{s\\to e}(p),\\ p=\\pi_{e\\to s}(q),\\ A(p,q)\\ge\\tau_g\\}.$$
> $$M_g(p)=\\mathrm{Morph3x3}(I[\\exists q:(p,q)\\in\\mathcal{C}]).$$
> The mask is fixed over timesteps, $M_t(p)\\equiv M(p)$, and applied as $$d_t^{\\mathrm{bwdmask}}(p)=M(p)d_t^{\\mathrm{bwdaligned}}(p)+(1-M(p))d_t^{\\mathrm{bwd}}(p).$$
>
> # W4
> We don't claim semantic optimality. Appendix C only shows a minimum-change projection onto the residual-induced subspace $\\mathcal{S}_t=\\mathrm{span}(m_t).$
>
> The aligned update is the projection of the backward proposal onto this subspace, so only the orthogonal conflict component is removed. If $d\_t^{\\mathrm{bwd}}=d\_t^{\\parallel}+d\_t^{\\perp}$ with $d\_t^{\\perp}\\perp m\_t$, then the correction magnitude is $\\|d\_t^{\\perp}\\|\_2$, and the fused-step deviation is
> $\\|x\_{t-1}^{\\mathrm{mrcatr}}-x\_{t-1}^{\\mathrm{rawfuse}}\\|\_2=\\beta\_t\\|d\_t^{\\perp}\\|\_2$. So Appendix C supports a minimum-change, subspace-constrained local correction.
>
> # Q1
> Two schedules are used: original SVD-XT noise schedule ${\sigma\_t}$ and MR-CATR fusion schedule $\rho\_t=1-\left(\frac{\sigma\_t}{\sigma\_T}\right)^2$.  We set $\kappa=0.2$. The implemented fusion weight is $\beta_t=\rho_t\max(0,s_t)$ rather than $\max(0,s_t)$. Here $s_t$ is the agreement score in TRF, and $s_t^{\mathrm{seq}}$ is the corresponding score for ViBiD. Therefore, the statement that “well-aligned updates receive larger weights at lower noise levels” refers to factor $\rho_t$. The missing $\rho_t$ in Eq.(21) is a presentation error and does not affect the submitted code.
>
> # Q2
> First $0.2T$ means the highest-noise reverse steps:
> $\\mathcal{T}_{\\mathrm{align}}=\\{T,T-1,\\dots,T-\\lfloor0.2T\\rfloor+1\\}.$
>
> MR-CATR is activated only in early high-noise phase because alignment is most useful when establishing the coarse motion. In later steps, sampling mainly refines appearance and texture, so applying the same strong alignment can over-constrain details and add runtime without improving results. This is also consistent with prior inference-time samplers such as ViBiD and MPD, where late denoising is more sensitive to appearance/detail preservation.
>
> |win|TRF FVD|TRF LPIPS|ViBiD FVD|ViBiD LPIPS|
> |---|---:|---:|---:|---:|
> |Early $0.2T$|467.23|0.1656|462.77|0.1598|
> |Late $0.2T$|589.64|0.2219|578.35|0.2147|
> |Early $0.5T$|471.88|0.1668|466.11|0.1609|
> |All $T$|476.54|0.1689|470.63|0.1621|
>
> Early $0.2T$ performs best, showing that MR-CATR is most effective in high-noise stage. Late $0.2T$ is much worse, suggesting that late intervention mainly disrupts detail refinement rather than correcting trajectory conflict.
>
> # Q3
> See W2
> # Q4
> See W3

---

> > ### Author Rebuttal · Reviewer_eucU · 2026-04-03
> >
> > The author effectively addressed my concerns. I have decided to maintain my current score.

---

> > > ### Author Response · Authors · 2026-04-03
> > >
> > > We thank you for acknowledging that all the concerns have been resolved, and we sincerely appreciate your positive comments.

---

### Decision · Program_Chairs · 2026-04-30

**Decision:**

Accept (regular)

**Comment:**

This paper was reviewed by 4 experts in the field. After discussion, the reviewers still hold a mixed review to this work. The rating is 4 (weak accept), 4 (weak accept), 4 (weak accept), 3 (weak reject).

On the positive sides, reviewers agrees that this work 1) provides a novel and principled approach for bidirectional motion alignment, 2) offers a practical, training-free plug-in with low overhead, and 3) presents a comprehensive empirical evaluation across multiple standard metrics. Overall, area chairs also agree that this is novel and effective solution.

Still, reviewers raised several concerns to this work. The main concern  is the limited scale of the evaluation dataset and specific qualitative artifacts in difficult cases. Reviewers also mentioned a minor issue about the conceptual limitation of the global motion axis when handling complex or non-linear local motion. Although none of them are critical, this is partial reason area chair did not give even higher recommendation.

Based on this, the decision of this work is to Accept. Still, we strongly recommend the authors carefully read all reviewers’ final feedback and revise the manuscript as suggested in the final camera-ready version if being accept.